

# Air pollution near arterial roads: An experimental and modelling study

José Ignacio Huertas Cardozo, Daniel Fernando Prato Sánchez

Energy and climate change research group, Tecnológico de Monterrey, Monterrey, 64849, México

*Correspondence to*: José Ignacio Huertas (jhuertas@itesm.mx)

**Abstract.** Aiming to advance in the understanding of pollutant dispersion near arterial roads, we measured, simultaneously, meteorological parameters, emission mass rates and TSP, PM10 and PM2.5 concentrations at several locations downwind two roads, located on a flat region without any other source of pollutants. We also implemented on a state of the art commercial CFD software, an air quality model to simulate the dispersion of solid and gas-phase pollutants emitted from

arterial roads. Numerical results of long-term averages and daily measurements of particle concentration showed high correlation with experimental measurements (R2>0.76).

We found that the plots of pollutants concentration vs distance to the road edge describe a unique curve when expressed in terms of non-dimensional numbers and that this curve is well described by a beta function. Profiles of vertical concentration sketch an exponential function at the road edge, an S shape downwind and a flat shape far from the road. Particles exhibit a

Rosin Rambler size distribution with average diameter of ∼ 7 µm. This distribution remains unaltered downwind from the road, which implies that at any location, PM10 and PM2.5 concentrations are a constant fraction of TPS concentration. Experimental data confirmed this observation.

Previous results can be used to determine the size of the area impacted by roads, identify mitigating and adaptive countermeasures, and to improve the accuracy of vehicular emission factors.

## 1.  Introduction

Land transportation is essential to the world's economy. This sector is the largest energy consumer and the largest emitter of air pollutants. Transport is responsible for around half of all energy related  nitrogen oxide emissions (56 Mt in 2015) and is an important source of primary particulate matter (International Energy Agency, 2016). As a direct consequence, people

living near roads are exposed to high concentration levels of vehicular emissions (CO, NOx, SOx, VOCs, and particles) (Parrish, 2006). To protect nearby population, environmental regulators must first quantify the size of the impact area affected by the use of roads and then identity preventive and mitigating countermeasures. The experimental determination of the road impact area is expensive and requires long-term (∼ 1-year) measurements of pollutant concentration at several locations downwind from the road. The identification of countermeasures requires a deep knowledge of the pollutant

dispersion phenomena near roads.



Near-road air pollution can be classified in i.) Street canyons formed by blocks of buildings in urban centers, ii.) Street intersection where pollutant concentrations are believed to be the highest and iii.) isolated roads such as urban arterial roadways or rural highways where 95% of traffic happens (Vardoulakis et al., 2003). We concentrate on arterial roads in consideration of their geometric simplicity and relevance.

5 Multiple publications have reported simultaneous measurements of pollutants concentration at several distances from arterial roads and at several heights, along with meteorological parameters and vehicular traffic. Before 2000, only few experimental works had been completed (Roorda-knape et al., 1998; Chock, 1978). They focused in the measurements of non-reactive pollutants (particles and CO) and used them to calibrate Gaussian-type line-source dispersion models. Those studies reported that pollutant concentration reduces exponentially with distance to the road edge and that the gradient of reduction became 10 negligible within a distance of ~0.5 km. The number of experimental works increased with the advance to technology to measure pollutants concentration. Some works concentrated in the determination of vehicle emission factors for unpaved roads through the determination of vertical profiles of concentration near the diffusive source and then the calculation of vertical and horizontal mass fluxes of particles (Veranth et al., 2003; Gillies et al., 2005; Zhu et al., 2002). High uncertainty in their results was attributed to the unidentified vertical concentration profile that could not be determined with the limited 15 number of available experimental data points. Some other examined particle size distribution of ultrafine particles near diesel prevalent highways and observed a multimodal distribution that changed as function of distance to the road and season of the year (Zhu et al., 2002; Baldauf et al., 2009). Others focused on the NO/NO2/NOx/O3 system near suburban areas and observed that their concentration reduce exponentially with distance except O3 that increases it concentration with distance as result photochemical reactions (Baldauf et al., 2007; Clements et al., 2009; Chaney et al., 2011). Most recently, an 20 important number of works focused on VOCs, especially benzene, butadiene and toluene that are recognized for being motor vehicle tracers (Venkatram et al., 2009). All these studies are limited in the number of sampling locations (<5) and in the time extension of their measurements (~days).

Air pollution near roads can also be studied by modelling the dispersion of the pollutants emitted from roads. Extensive work has been conducted to develop accurate models to predict pollutant concentrations near roads. The vast majority of them are 25 Gaussian-type models. Currently, the USEPA recommend the use of AERMOD for this purpose as full replacement of CALINE4 (USEPA, 2015), which has been the preferred model of the environmental community. Previous studies have shown good performance of these models reproducing long term and peak concentrations of particulate matter across the road direction (Hagler et al., 2010). However, they present problems predicting pollutant concentration for short periods (< ~1 day). They do not consider the physical properties of the pollutants such as density or diffusivity and therefore predict the 30 same concentration at a given location for particles that for CO or NOx. Some studies have reported problems matching experimental data of gaseous pollutants such as CO (Sharma et al., 2013). These models also over predict particle concentration for cases with low winds speed (<1 m/s) or wind directions nearly parallel to the road (H. Chen et al., 2008; Holmes and Morawska, 2006). Finally, these models cannot be used to gain insight into the dispersion phenomena occurring near roads because they are heuristic models (assume that pollutants disperse following a normal distribution in the 35 horizontal and vertical axis), instead of solving the physic-chemical equations that describe the dispersion of pollutants in the atmosphere (mass conservation, momentum, energy conservation, chemical reactions, and aerosol dynamics).

Several authors have attempted to solve those equations numerically and have named their models as Eulerian or Lagrangian models depending on the CFD technique they implemented. Computational fluid dynamics (CFD) is a branch of fluid mechanics that uses numerical methods to solve the differential equations that describe the flow of fluids over or through 40 solid bodies. CFD techniques can be used to solve the equations that describe the dispersion pollutant in the low troposphere and to model the transport of solids in the form of gas-phase suspended particles, and their deposition. Recent advances in computing technology have made available to the public reliable CFD tools that were limited for researchers from other disciplines, with high computational capabilities.

Few works have used commercial CFD tools in connection to pollution near urban and rural roads. There is not any 45 established "air quality CFD model" yet, at least for the study of air pollution near roads. Existing reported works have focused more on the study of the emission source rather than on the dispersion of pollutants downwind the road. Sahlodin et



al., (2007) modeled via CFD the vehicle-induced turbulence (VIT) on a paved road and incorporated their results into a Gaussian dispersion model to study the dispersion of CO. Wang and Zhang, (2009) advanced this work and added the turbulence generated from the road (RIT) in their model. Comparing their results against experimental data and CALINE4 they obtained better predictions for short time intervals of CO concentrations near two highways. Chen et al., (1999), Tong
et al. (2011) and Gerardin et al. (2016) used CFD to study the dust patterns originated by vehicles moving over desert terrains. They did not study the dispersion of those particles downwind.

To advance in the understanding of pollutant dispersion near roads, we implemented on a state of the art commercial CFD software, an air quality model, and calibrated it by comparing its results against measurements of downwind pollutant concentrations. Then, we systematically used this model to determine concentration profiles downwind in the horizontal and
vertical direction as function of meteorology parameters and emission rates. This paper focuses on the use of the numerical model to gain insight into the pollutant dispersion phenomena near roads. A companion paper concentrates on the description of the model (Huertas et al., 2017). This manuscript reports the following contributions:

- We report simultaneous measurements of meteorological parameters, mass emissions and concentrations of TSP, PM10 and PM2.5 at five locations near two unpaved roads located on flat terrains with no obstacles for pollutants dispersion,
nor additional sources of particles.
- We implemented, on a state of the art CFD software, a model to simulate the dispersion of solid and gas-phase pollutants near arterial roads, that resolves the issues of the Gaussian models with low wind speeds and wind directions nearly parallel to the road, and whose results are highly correlated with short and long-term experimental measurements of particle concentration.
- We report that solid and gas-phase pollutant concentrations vs distance to the road sketch a beta function, and that when those concentrations are expressed in terms of non-dimensional parameters for concentration ($C^* = C\ U\ Sc\ /E$) and distance to the road ($x^* = x/L$.), they collapse into a single curve that depends only on the phase of the pollutant.
- We report that particles near arterial roads exhibit a Rosin Rambler size distribution that remains unaltered downwind and that at any location, PM10 and PM2.5 concentrations are a constant fraction of TPS concentration.
- We determined the size of the environmental impact area generated by the use of roads.

## 2. Methodology

Aiming to study pollutant dispersion from arterial roads, we applied the experimental design illustrated in Figure 1.

We selected a region in which the only particle emission source was the road. This assumption holds true for areas covered with vegetation, such as pastures. Then, we selected roads of general characteristics, i.e., roads located over flat terrain and
without the presence of obstacle for the dispersion of pollutants. Considering the relevance of unpaved roads as source of air pollution and the advantages that they offer to study dispersion phenomena, we selected unpaved roads for this study. Pollutant emissions on roads come from vehicle exhaust pipes, tire wear and particle resuspension due to wind erosion and to the action of the tires on the road. In the case of unpaved roads, particle emissions due to resuspension are at least four orders of magnitude higher than the other sources. This abundance in the amount of pollutant being emitted facilitates the
experimental work and learning process of the dispersion phenomena. Furthermore, unpaved roads are currently of great interest. Unpaved road dust is the highest single emissions category within the non-point fugitive dust category, accounting for about one third of non-windblown fugitive dust emissions in the USA (Pouliot et al., 2012).

We measured road characteristics along with daily pollutant concentrations at different distances from the road, meteorological variables, and vehicular traffic, over prolonged periods (~ 1 month). The extent of field campaigns was
limited to available financial resources.





Then, we implemented on AERMOD and a commercial CFD software, models to simulate the dispersion of pollutants under the conditions measured during the monitoring campaign. We compared measured concentration profiles with the corresponding simulation estimates.

We used the CFD calibrated model to study the effect of particle emission rate, pollutant physical properties, road width, and
meteorological parameters on pollutant concentration and particle size distribution downwind from the road edge. We studied the concentration profiles downwind from the road, in the horizontal and vertical direction. Finally, we used this model to determine the size of the impact area generated by the experimental roads monitored in this study.

### 3.   Experimental work

*Roads:* We selected as cases of study two sections of road created by petroleum industry projects in Eastern Colombia. Both roads are located in a tropical zone, on flat terrain, with plains covered by pasture and without trees. The first road section, labeled *Met*, is 213 m above sea level and 11.5 m wide. The second section of road, labeled *Cas*, is 172 m above sea level and 8 m wide. The two road sections are 136 km apart.

*Road´s silt content:* We took random samples of road material. We determined road surface silt content using method INV E
123 from the Colombian National Road Institute (Instituto Nacional de Vías, 2007). Silt content is the mass fraction of particles with diameters smaller than 200 µm present in the road surface material. We found that both roads had a silt content of 9.5%.

*Traffic:* During the measurement campaign, we registered the average speed of each vehicle and its size using the following categories: motorcycle, automobile, pick-up truck, utility vehicle, bus, tanker truck, two-axle truck, and three-axle truck.
Figure 2.a shows vehicle distribution by size for road *Cas*. For this road, average vehicular flow was 20.9 vehicles/h, with an average weight of 15.8 ton and an average speed of 25 km/h. For road *Met*, these values were 47.2 vehicles/h, 27.6 ton, and 25 km/h, respectively.

*Particle emission rate* from the road cannot be measured directly, given their fugitive nature. Present methods of estimating this variable for use in dispersion models are based on emission factors. The USEPA (1998) published emission factors ($E_f$)
for unpaved roads. Particle mass emission rate ($E$) was obtained using Equations 1–4 (USEPA, 2006; Huertas et al., 2012a) where symbols are defined in the symbols and acronyms table. $E$ includes tail-pipe, tire wear and resuspension emission sources. Using Equations 1-4, we estimated average TSP emissions for roads *Cas* and *Met* of 49.2 and 99.3 g/s-m$^2$, respectively.  Similarly, we obtained PM$_{10}$ emissions of 14.39 and 29.03 g/s-m$^2$ for roads *Cas* and *Met*, respectively.

$$E = \frac{\sum N_i E_{fi}}{3.6\,L}(1 - \eta_s)(1 - \eta_r) \tag{1}$$

$$E_{fi} = k\left(\frac{s}{12}\right)^a\left(\frac{W_i}{3}\right)^b \tag{2}$$

$$\eta_s = \frac{0.8\,p\,r\,t}{q} \tag{3}$$

$$\eta_r = \frac{n_d - m}{n_d} \tag{4}$$



*Meteorology:* We measured primary and secondary meteorological variables (wind speed, wind direction, radiation, precipitation, humidity) using a meteorological station meeting the World Meteorological Organization (WMO) criteria (WMO, 2008). Figure 2.b shows the wind rose obtained for road *Cas*. No rain fell during our measurement campaign. High temperatures and humidity conditions (average 31.6ºC and 75% respectively) were typical of summer in this tropical zone.

*TSP, $PM_{10}$ and $PM_{2.5}$ concentration downwind from the road edge:* We used four high-volume samplers meeting USEPA recommendations to measure TSP and $PM_{10}$ concentrations (USEPA, 1999). Observing preferential wind directions, we located the samplers downwind at distances of 5, 60, and 125 m from the road edge (Figure 1). We also located a sampler upwind, at a distance of 5 m from the road edge. TSP and $PM_{10}$ concentrations were determined by weight differences of quartz filters exposed to air flow for 24 h in the samplers (USEPA, 1999). Concentrations of TSP, $PM_{10}$ and $PM_{2.5}$ were also
determined with two automatic stations, which measured particle concentration through a nephelometer with a resolution of 0.1 µg/m$^3$. They were located 5 m upwind and 5 m downwind. Results were expressed per local air quality standards, in terms of daily concentrations, and geometric average over the measurement campaign intervals. Results will be shown in Figure 5.

*Particle size distribution:* We determined particle size distribution of airborne particles using the methodology reported by
15 Huertas et al. (2012b). We measured observable diameters of a random sample of particles trapped in the TSP sampler filters described above, at different magnifications using a JEOL NeoScope JCM-5000 scanning electron microscope (SEM). Figure 2.c shows the particle morphology. Figure 2.d shows that, independent of the distance to the road edge, TSP had a log-normal Rosin-Rammler distribution ($Y_d$), described by Equation 5, with an average diameter $\bar{d}$ of 7.08 µm, scatter parameter $n$ of 1.27, and pre-exponential factor $Z$ of 0.152 for road *Met*. For road *Cas*, the parameters were $\bar{d}$ = 5.70 µm, $n$ =
1.23, and $Z$ = 0.191.

$$Y_d = Z\,e^{-\left(\frac{d}{\bar{d}}\right)^n} \tag{5}$$

*Particle composition:* We used SEM coupled with energy-dispersive x-ray spectrometry (SEM-EDS) to determine composition of particles trapped in the TSP sampler filters described above. In this study, a JEOL JSM-6360- LV and a JSM 6510 LV microscope were used. The microscopes were operated at an accelerating voltage of 20kV at low vacuum (LV).
Figure 2.e is a graph of the SEM–EDS results for a sample before and after collection. Regardless of the sampler location, or the date, C, Si, and O were the main elements present (~89% of the mass concentration). Via X-ray photoelectron spectroscopy technique (XPS), we confirmed that C, O, and Si are the main elements present in the particles with average mass concentrations of 41.5%, 34.7%, and 5.7%. The particles also contain large amounts of K (11.6%), as well as an average of 2.0% Zn and 1.9% Mg and traces of Al, Na, Ba, S, Fe, and Cr.

## 4. Modelling pollutants dispersion near roads

We implemented on AERMOD and on a state of the art CFD software, air quality models to study the dispersion of pollutants near arterial roads and to assess their environmental impact. We will refer to them as the AERMOD model and the NR-CFD model (Near roads air quality CFD model). This section summarizes the implementation of these two models.
Further details on their implementation are reported in (M. E. Huertas et al., 2017; Huertas et al., 2017).



### 4.1 AERMOD model

We used the topographical, meteorological and emissions data described above as input data for AERMOD and simulated
the dispersion of TSP near roads Cas and Met. We used 13 km long roads located over flat terrains. Following the USEPA
recommendations, we modeled the roads as area sources (USEPA, 2003). Figure 3 shows that concertation isopleths are
parallel to the road and that particle concentration decreases with the distance to the road.

### 4.2 NR-CFD model

Particle dispersion is a natural phenomenon that varies with time. To determine its impacts on human health and
the environment, average short-term (~1 day) and long-term (~ 1-year) ground-level concentrations are needed.
As the modelling of transient-state particle dispersion via CFD for 1 day or 1 year is computationally prohibitive,
we simplified the problem by using short-interval modelling, where a steady state condition can be assumed. In
practice, 1-hour intervals are appropriate, as meteorological data are reported in this way.

*Topography*: Figure 1 shows the computational domain used in this study. We used a 1500-m-long, 25-m-high
and 10 m-depth domain.

*Meteorology:* Besides solving the physics equations governing fluid flow and heat transfer, the simulation of
atmospheric dynamics is achieved in CFD through the specification of boundary conditions. We used a condition
of symmetry (zero gradient normal to boundary) at the upper ceiling. Pressure outlet was used as boundary
condition at the exit and a periodic boundary condition was used for the lateral walls. On the surface downwind
from the road, we considered the *Air–particulate matter–ground interaction* and used the boundary condition that
traps the particles arriving at the surface (ANSYS, 2012a). We expressed the entry of air into the computational
domain as a speed profile of a fluid on a flat surface using Equation 6, which describes a neutrally stratified
atmospheric boundary layer (Panofsky and Dutton, 1984; Zanneti, 1990),

$$u = \left(\frac{u^*}{K}\right) \cdot ln\left(\frac{z+z_0}{z_0}\right), \tag{6}$$

where $u$ is the wind speed at a height z, $K$ is the universal Von Karman constant with a value of 0.4, $u^*$ is the
friction velocity obtained experimentally (we used values of 0.06–0.52 m/s), and $z_o$ is the surface roughness,
which depends on soil use. For vegetation-covered pasture-type soils, the USEPA recommends using a value of
0.3 m (2013). For simplicity, we ignored the effect of heat transfer between the atmosphere and the Earth's
surface on particle dispersion. Finally, we selected the standard $k$-$\varepsilon$ turbulence model, which is based on models
of kinetic energy transport equations ($k$) and dissipation rate ($\varepsilon$) (Launder and Spalding, 1983).

*Mass emission of pollutants:* The emission rate *(E)* includes tail-pipe, tire wear and resuspension emission
sources. Road emissions are primarily silicates that are inert to elements present in the atmosphere (Huertas et al.,
2012b). For our simulations, we selected quartz, a type of silica with a density of 1730 kg/m³, as the particle
material. To uncouple model results from uncertainties on the emission data, we selected an arbitrary value of 1
g/s-m² over the total road area and reported results per emission unit. To compare model results with
experimental data, we specified that particles incoming into the computational domain exhibit a Rosin-Rammler
size distribution ($Y_d$), with the parameters obtained experimentally and described in section 3.



*Model setup:* We did not consider chemical reactions nor the hygroscopic effect of particles in high-humidity environments. We used quad-type elements to discretize the computational domain showed in the Figure 1. We applied a structured mesh because it is easy to implement, requires less computing time, and facilitates particle tracking. Vertical grid refinement was carry out. The height of the first cell was 0.06m, which is equivalent to a
$y+$ value of 214.2. This value satisfies the log law of modelling that recommends $50 < y+ < 300$ (Blocken et al., 2007). We used 3.4 million of computational cells for the simulations with refinement in the regions of interest, such as the sections adjacent to and downwind from the road. This mesh was selected after a grid independence analysis from 0.11 to 3.9 million elements. On a server with 16 parallel processors, solution times ranged from 30 min to 3 h for the finest mesh. We used the double precision solver of ANSYS FLUENT v.17, as
recommended for large geometries and significant pressure and speed variations (ANSYS, 2012b).

*Post-processing:* We estimated particle dispersion for 14 typical wind speeds in the range of 0.25–5.5 m/s. Results were reported in terms of non-dimensional parameters of concentration ($C^*$), distance to the road edge ($x^*$) and speed ratio ($U^*$), defined by Equations 7, 8, and 9, respectively. In all cases, reported concentrations were obtained at ground level.

$$C^* = \frac{C\,U\,Sc}{E} \tag{7}$$

$$x^* = \frac{x}{L} \tag{8}$$

$$U^* = \frac{U}{w_{po}} \tag{9}$$

where $C$ is particle concentration, $w_{po}$ is particle emission speed in the vertical direction, $L$ is road width, $x$ is the
distance from the road edge, $U$ is wind speed, and $Sc$ is the Schmidt number. For the dispersion of particles $Sc=1$.

To obtain pollutant concentrations over extended periods, we calculated for each hour $i$ the values of input parameters for the NR-CFD model. Then, pollutant concentration ($C_{i,j}$) is obtained for each hour and position ($j$) downwind from the road. Finally, average daily and annual values are obtained for each distance from the road ($\overline{C}_J$). If particle emission is held constant, average values are obtained by Equation 10 where $f_{k,q}$ is the frequency
at which speed $U_{k,}$ appears in the wind rose for each wind direction ($q$).

$$\overline{C}_J = \sum \sum f_{k,q}\, C_{k,j} \tag{10}$$

For winds flowing in directions other than perpendicular to the road, we maintained the magnitude of wind speed
unaffected and computed its contribution to particle concentration at receptor $j$ as if the receptor $j$ were located at an equivalent distance from the road ($x_e$) (Figure 4, Equation 11).

$$x_e = x / Cos(\theta) \tag{11}$$





*Calibration of the NR-CFD model:* we used the implemented NR-CFD model to simulate the conditions observed for particle dispersion during the two monitoring campaigns. Then, we obtained daily and long-term average values of TSP and PM2.5 concentration downwind from the road using the procedure outlined above, and compared them with experimental measurements.

Figure 5.a shows high correlations between daily TSP concentration results from the NR-CFD model and experimental data ($R^2$ =0.82, 0.76, 0.85 and 0.83 at -5, 5, 60 and 125 m downwind, respectively) for road *Cas*. A similar result was obtained for road *Met* where we obtained $R^2$ =0.92, 0.82, 0.98 and 0.98 at -5, 5, 60 and 125 m downwind, respectively. These results demonstrate the capacity of the NR-CFD model to reproduce short term measurements.

Similarly, Figure 5.b shows that long-term averages of NR-CFD results correlate well with experimental data ($R^2$ = 0.76 for *Cas* and 0.98 for *Met*). In this figure, each data-point represent the average value of more than 15 data points of TSP concentrations. This figure also shows that for the NR-CFD model results to reproduce the experimental data, the model must be calibrated by applying a calibration factor to correct errors in particle mass emission estimation ($E$) and particles emission speed ($w_o$). It shows that the calibration factors were 0.16 and 0.70

for roads *Cas* and *Met*, respectively.

Using previous adjustments, we obtained a fractional bias *FB*= 4.85E-04 and a root mean square error *RMSE*= 41.46 for road *Cas* and *FB*= 3.10E-03 and *RMSE*= 58.97 for road *Met*. These metrics are typically used to evaluate the performance of dispersion models (Huertas et al., 2012). *FB* ranges between -2 to +2, evaluates the sub or over estimation of the model and *FB*=0 is the best. *RMSE* provides information on the deviation of the

model, ranges from 0 to ∞ and *RMSE*=0 describes best models.

## 5   Use of the AERMOD and NR-CFD models to study particle dispersion near arterial roads

Effect of variations in pollutants emissions rate on pollutant concentrations downwind: We varied emission rate to 0.1, 10

and 100 times the initial base rate (Eb = 1 g/s m2). The AERMOD model showed that the long-term average non-dimensional concentration (C*) as a function of non-dimensional distance to the road (x*) remained the same (Figure 6.a). Figure 6.b shows the results obtained by the NR-CFD model. Again we observed that the profile of C* vs. x* remained the same. This result implies that the pollutant concentration downwind from the road is proportional to the emission rate.

Effect of wind speed:  For the case of AERMOD we explored the effect of wind speed on pollutant dispersion varying the

meteorological conditions. Data were derived from highly diverse geographically regions. Five data sets came from monitoring sites in the USA (WebMet, 1990), three from England (BADC, 2012), and four from Colombia (IDEAM, 2009). We found that the concentration profiles obtained were the same when expressed in terms of C* vs x* (Figure 6.c).  In the case of the NR-CFD model, Figure 6.d shows TSP concentration as function of distance to the road as wind speed changes. Again, we observed that the concentration profile shape remained similar. This observation implies that TSP concentration

downwind is inversely proportional to wind speed, which agrees with several other authors' observations (He and Dhaniyala, 2012), but contrary to people's expectations.

*Comparison of the AERMOD and NR-CFD models results:* AERMOD and NR-CFD results showed that non-dimensional pollutant concentration at ground level decreased with distance to the road edge following a beta function. We found that

$\alpha_1$=0.344, $\alpha_2$=1.17 and $F_d$=34.99 were the beta function's parameters that best fit AERMOD estimates of TSP concentration with $R^2 > 0.95$. In the case of the NR-CFD model simulating TSP dispersion, the beta function's parameters were $\alpha_1$=1, $\alpha_2$=484.2 and $F_d$=13.7, with $R^2 > 0.9$. We explored the effect of the parameters' differences on the long-term average concentrations predictions of both models. Figure 6.d shows average results over the campaign period. Both sets of results fit beta functions, but the concentration profiles differ, especially for concentration values close to the road, where the NR-CFD




model reproduces better than AERMOD experimental measurements. This result demonstrates that NR-CFD modelling, and the methodology proposed in section 4.2 which includes the contribution on particle concentration of winds blowing in directions nearly parallel to the road, resolve the issue of Gaussian models, which overestimates particle concentration in this area. We also compared AERMOD and NR-CFD results through a correlation analysis and as expected we found high levels

of correlations ($R^2 > 0.98$ for *Met* and $R^2 > 0.98$ for *Cas*). We did not conduct correlation analysis for daily values, as AERMOD produces unreliable results over short time periods (Zou et al., 2010).

*Particle size distribution downwind from the road:* Gaussian models, and in particular AERMOD, do not include particle size distribution. The NR-CFD model does. Figure 7.a shows that the particle size distribution downwind from the road, for

particles with $d < 30$ □m, remains constant and exactly equal to the input particle size distribution. This result agrees with experimental results shown in Figure 2.d. However, this result was unexpected as particles residence time in the atmosphere depends directly on the settling speed. The Stokes settling equation estimates the settling speed ($V_s$) of particles in a fluid for $Re < 10$ as follows (King, 2002):

$$V_s = \frac{g_z\,(\rho_p - \rho)}{18\,\mu}\,d^2 \qquad (12)$$

where each variable is defined in the nomenclature table. According to Equation 12, $V_s$ depends on the square of particle size, and therefore larger particles are expected to be deposited at shorter distances affecting particle size distribution downwind. However, for particles with $d < 30$ □m dispersing in atmospheric air, $V_s$ is small ($V_s < 1$ cm/s) and therefore the

settling residence time ($t_s \sim 15$ min) is larger than the residence time of the particles within the computational domain ($t_r \sim 5$ min). Then, changes in particle size distribution are negligible within ˜1 km from the road edge.

*$PM_{10}$ and $PM_{2.5}$ concentrations downwind:* Previous result implies that $PM_{10}$ and $PM_{2.5}$ concentrations at any location $< 1$ km downwind from the road is a constant fraction (*f*) of TSP concentration. The values for these fractions depend on the input particle size distribution. Using Equations 13 and 14 and NR-CFD results, we obtained $f_{PM10}$=85% and $f_{PM2.5}$=34% for

road *Met* respectively. For the case of road *Cas* we obtained $f_{PM10}$=90% and $f_{PM2.5}$=41%.

$$f_{PM10} = \frac{\int_0^{10} Y_d\ dz}{c_{TSP}} \qquad (13)$$

$$f_{PM2.5} = \frac{\int_0^{2.5} Y_d\ dz}{c_{TSP}} \qquad (14)$$

To confirm previous results, we compared simultaneous daily measurements of TSP, $PM_{10}$ and $PM_{2.5}$ concentrations. Figure 7.b shows high correlation levels ($R^2 > 0.89$) of TSP with $PM_{10}$ and of TSP with $PM_{2.5}$ in the measurements of the automatic stations located 5 m upwind and 5 m downwind road *Cas*. This figure also shows $f_{PM10}$=83% and $f_{PM2.5}$ =6%, which agrees with previous results.
We also compared simultaneous daily measurements of TSP and $PM_{10}$ obtained by high-volume samplers located 5 m

upwind and 5, 60 and 125 m downwind this road and again obtained a high correlation level ($R^2 > 0.89$), but in this case we obtained $f_{PM10}$=56%. These experimental results confirm the observation of constant particle size distribution downwind from the road, but leave unclear the values for $f_{PM10}$ and $f_{PM2.5}$.





*Vertical profiles of particle concentration:* Frequently air quality models report results of pollutant concentration at ground level and those results are contrasted with measurements obtained with sensors located at few meters (~2 m) above ground level. This practice assume that pollutant concentration remains constant with height. Additionally, researches have looked at differences in pollutant concentrations in the vertical direction, near diffusive sources, aiming to calculate vertical fluxes and then estimate the corresponding mass emission from the diffusive source (Etyemezian et al., 2004). Finally, pollutant concentrations are compared with NAAQS to assess their impact on human health. However, people can be exposed at very different pollutant concentration levels depending on the height respect to the ground level where they usually live. Therefore, there is great interest in observing variations of pollutant concentration with height.

We observed the concentration of pollutants in the vertical direction at several distances downwind from the road (Figure 8). Intuitively, we expected that the AERMOD model showed a vertical Gaussian profile centered at any height above ground level. However, it showed (Figure 8.a) that peak values of average particle concentration, at any location downwind, always occur at ground level, which corresponds to the height of emission source. At the edge of the road, particle concentration was the highest. This concentration reduces exponentially with height and becomes negligible at ~15 m. Downwind from the road, the vertical profile does not expand in the vertical direction reflecting a negligible diffusion effect. Thus, this vertical exponential profile is preserved except near ground level where deposition reduces particle concentration making this profile flat. With distance from the road it becomes essentially flat. We highlight that AERMOD was not originally designed for the observation of vertical profiles and that it is a heuristic model, therefore, these results do not necessarily correspond to reality.

Figure 8.b shows NR-CFD results. Similar to the AERMOD model, the vertical concentration profile exhibits an exponential shape at the road edge. However, within the first meter downwind from the road, the vertical concentration profile exhibits a sharp "S" shape that broadens vertically with distance to the road edge, as result of particle deposition at ground level and vertical diffusion. Peak values occur at intermediate heights that grow with distance to the road edge. At $x\sim$ 100, this peak reaches ~1.75 m, which is the average peoples' height. Far from the road edge (~1000 m), this profile becomes essentially flat. These observations agree with the qualitative description of the vertical profiles reported by Etyemezian et al. (2004) and experimental results (Chaney et al., 2011).

*Horizontal mass flow of pollutants*: Several authors have determined the mass emitted, per unit length, of a pollutant from a diffusive line source through measurements of its horizontal mass flow ($\dot{m}_{H,i}$) downwind the source and affecting this results by a depletion factor. Equation 15 calculates the horizontal mass flow of pollutant $i$ at a given distance from the road.

$$\dot{m}_{H,i} = \int u\, C_i\, dz \qquad (15)$$

where $u$ and $C_i$ are the wind speed and pollutant concentration at height $z$, respectively. We used previous results of wind speed and vertical profile of particle concentration, and calculated the depletion factor as the ratio of the horizontal mass flow and emission rate. Figure 8.c shows that this ratio is near to one at the road edge and decreases exponentially with distance from the road. At ~10$x^*$ it reaches 50% of emission rate and at ~20$x^*$ it reaches a stable value of ~40% of the emission rate. This result implies that at distances greater than ~20$x^*$ particle deposition is negligible and therefore ~40% of the mass emitted remains in the atmosphere. i.e. ~40% of the mass emitted is transported for very long distances. This result also implies that the emission rate from diffusive sources can be determined by measuring pollutant concentration at several distances from the source instead of measuring vertical concentration profiles, which is much easier to do.

*Dispersion of gas phase pollutants*: Gaussian models do not take into consideration the pollutants' physical properties related to mass transfer such as density, viscosity or diffusivity. Therefore, they produce exactly the same results of downwind concentrations regardless of the phase or physical properties of the pollutant under consideration. This result is clearly unrealistic.

Alternatively, we used the NR-CFD model to study differences in the dispersion of CO, $CO_2$, $NO_2$ and TSP. We observed that downwind pollutant concentration grows with diffusivity and reduces with viscosity. The inclusion of the Schmidt



number in the definition of the non-dimensional concentration takes into consideration differences in mass diffusivity and viscosity of pollutants. Thus, non-dimensional concentration of all gas phase pollutants exhibits a unique profile (Figure 9.a) that can be represented by a beta function with parameters $\alpha_1$=1.00 and $\alpha_2$=138.47 and $Fd$=55.73. As expected, at the road edge, solid and gas phase pollutant non-dimensional concentrations are similar. However, downwind, gas phase pollutants exhibit higher concentrations than solid phase pollutants because they do not experience sedimentation or scavenging downwind when they interact with the ground surface.

Aiming to validate these results, we looked in the literature for experimental data. Several works have reported measurements of gas phase pollutants concentration near roads. However, none of them reports simultaneous measurements of mass emissions, meteorological conditions and pollutants concentration that could be used to validate quantitatively the NR-CFD model. As an approximation, we compared, qualitatively, NR-CFD results to values of $NO_2$ measured near roads with high traffic of heavy-duty vehicles in UK (Chaney et al., 2011). Each set of simultaneous NO2 measurements were normalized in the way that the area under the concentration vs distance to the road edge curve were equals to one. Figure 9.b shows that numerical and experimental results are similar. Non-dimensional concentration differs from normalized concentration in their area under the curve but their shapes are similar.

We also attempted to compare NO and $O_3$ measurements reported in the same study with NR-CFD results. As expected, they were different. NO shows a much pronounced concentration reduction in the vicinity of the road edge a result of its rapid recombination into $NO_2$, while $O_3$ concentration increased with distance to the road edge as result of photochemical processes. The current version of the NR-CFD model does not include chemical reactions, therefore it cannot reproduce the concentration of pollutants like NO, and O3 that experience fast reaction rates near roads.

*Area affected by roads*: We defined the area affected by roads as the area near the road at which short or long term pollutant concentration surpass the threshold values recommended by the national atmospheric air quality standards (NAAQS). Those standards are level of exposure to pollutants considered harmful to public health and the environment (USEPA, 2012). Experimental and numerical results obtained with the AERMOD and NR-CFD models showed that pollutant concentrations decreases continuously with distance from the road. Therefore, the impact area was re-defined as the largest area parallel to the road limited by the distance at which the short or long-term concentration of any pollutant equal the corresponding threshold value specified in the NAAQS.

This work has shown that pollutants concentration downwind from the road is proportional to the mass emission rate and therefore the impact area scales with emission rate. Similarly, the impact area grows with the level of restrictiveness specified for the pollutant under consideration. Therefore, the ratio ($R_c$) of mass emission to NAAQS of a given pollutant (Equation 16) evaluates its level of restrictiveness. The pollutant with the largest $R_c$ defines the largest impact area and therefore it will represent the limiting scenario.

$$R_{c,i} = \frac{E_i}{NAAQS_i} \qquad (16)$$

In the case of transit on unpaved roads, $R_c$ for short-term $PM_{10}$, with a maximum allowed concentration of 50 µg/m$^3$ for 24 hr. exposure, shows the largest value of $R_c$ and therefore corresponds to the limiting case. Using the experimental data obtained during the measurement campaigns, and the calibrated results of the NR-CFD model, we obtained the impact areas for roads *Cas* and *Met*. They are the regions parallel to the roads within a distance of 73.2 m for road *Cas* and 549.7 m for road *Met*.





## 6    Conclusions

Aiming to advance in the understanding of pollutant dispersion near roads, we measured simultaneously meteorological parameters, emission mass rates and TSP, $PM_{10}$ and $PM_{2.5}$ concentrations at several locations downwind two unpaved roads located on a flat region without any other source of pollutants. We also implemented on a state of the art commercial CFD
software, an near road air quality model (NR-CFD model), and used the obtained experimental data to calibrate it.  We obtained high correlations ($R^2 > 0.76$ ) of the NR-CFD model results with long-term averages and daily measurements of particle concentration. We observed the NR-CFD model resolve the issue of Gaussian models overestimating pollutant concentration near the road edge and at low wind speeds.

We found that pollutant concentrations downwind describe a unique curve when plotted in terms of non-dimensional
numbers, independently of wind speed ($U$), emission rate ($E$) and the nature of the pollutant. The dimensional number for concentration is $C^* = C\ U\ Sc\ /E$ and for distance to the road is $x^* = x/L$. This result implies that pollutant concentration is proportional to $E$ and inversely proportional to $U$ and the Schmidt number ($Sc$), which compares the effect of viscosity and diffusivity in mass transfer processes. That curve can be approximated to a beta function of parameters $\alpha_1 = 1$, $\alpha_2 = 484.22$ and $Fd = 13.70$, for the case of solid phase pollutants and $\alpha_1 = 1$ and $\alpha_2 = 138.47$ and $Fd = 55.73$ for the case of gas phase pollutants.
Using micrographs of the particles trapped in receptors located downwind from the road, we observed that particle size distribution remained the same independently of the distance to the road edge. This distribution is well described ($R^2 > 0.99$) by a Rosin-Rambler function of parameters $Z = 0.191$, exponential factor $n = 1.23$ and mean diameter $d = 5.70$ µm for road $Cas$, and  $Z = 0.152$, $n = 1.27$ and $d = 7.08$ µm for road  $Met$. This observation implies that, at any location, the concentration of $PM_{10}$ and $PM_{2.5}$ is a fraction ($f$) of TSP. Experimental data showed that these fractions were $f_{PM10} = 85\%$ and $f_{PM2.5} = 34\%$ for road
$Met$, and $f_{PM10} = 90\%$ and $f_{PM2.5} = 41\%$ for road $Cas$. The NR-CFD model reproduced these results.

The NR-CFD model predicts that the vertical concentration profile exhibits an exponential shape at the road edge. Within the first meter downwind from the road, the vertical concentration profile exhibits a sharp "S" shape that broadens vertically with distance to the road edge, as result of particle deposition at ground level and vertical diffusion. Far from the road edge (~1000 m), this profile becomes essentially flat. This result agrees with experimental results reported in the literature.
We defined the road impact area as the bandwidth parallel to the road at which short or long-term pollutant concentrations surpass the threshold values recommended by the national atmospheric air quality standards (NAAQS). Using the experimental data obtained during the measurement campaigns, and the calibrated results of the NR-CFD model, we obtained that the impact areas for roads $Cas$ and $Met$ were within a distance to the road of 73.2 m and 549.7 m, respectively.

Results reported in this manuscript are useful to design actions of mitigation and adaptation for atmospheric pollution near
roads, and to improve the estimation of pollutant emissions near diffusive sources. However, results presented in this manuscript are limited to the case of dispersion of pollutant in atmospheres under neutral conditions without chemical reactions. Additional efforts are required to implement the effects of heat transfer and chemical reactions on pollutant dispersion near roads.

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





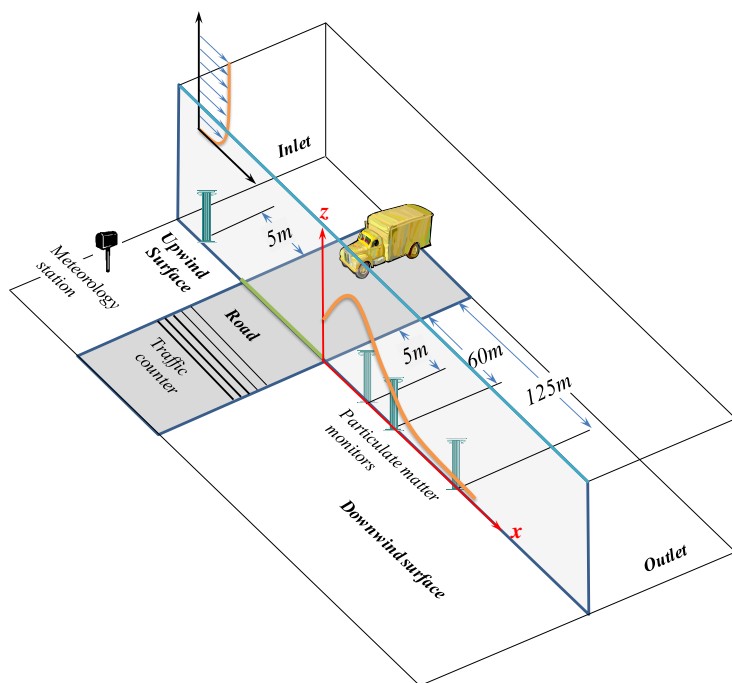

**Figure 1 Illustration of the experimental design and computational domain used for the study of pollutants dispersion near roads.**



*a*



*b*

*c*

*d*

*e*

**Figure 2 Experimental data obtained during the measurement campaign near road Cas. a) Average distribution of vehicle fleet circulating on road Cas. b) Wind rose. c) Morphology of particles in a high-volume sampler. d) TSP distribution obtained via SEM. e.) SEM- EDS results for quartz fiber samples before and after particle collection.**





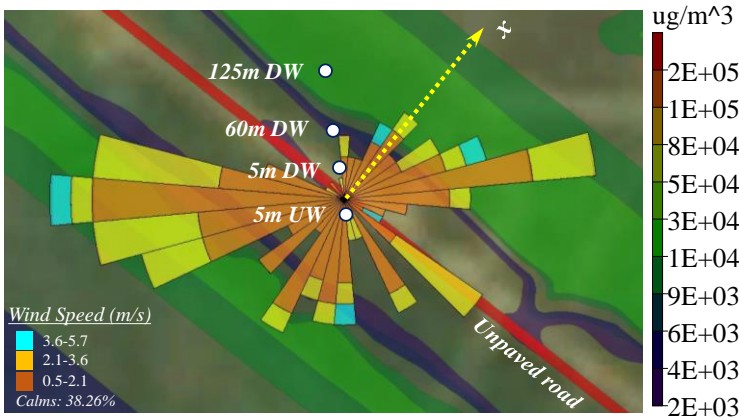

5    **Figure 3 AERMOD model results of average particle concentration near road Cas.**

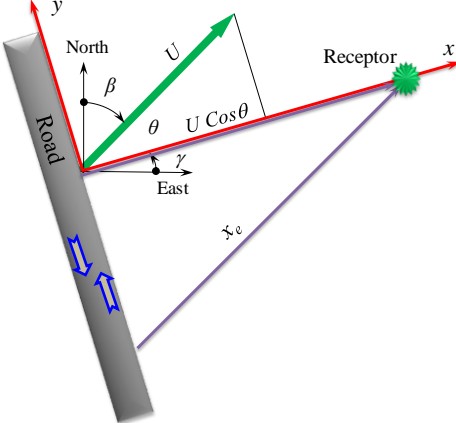

**Figure 4 Illustration of the methodology to include the contribution to particle concentration of receptors located downwind from the road, from winds blowing in directions ($\beta$) different to the angle of the line perpendicular to the road ($\gamma$).**



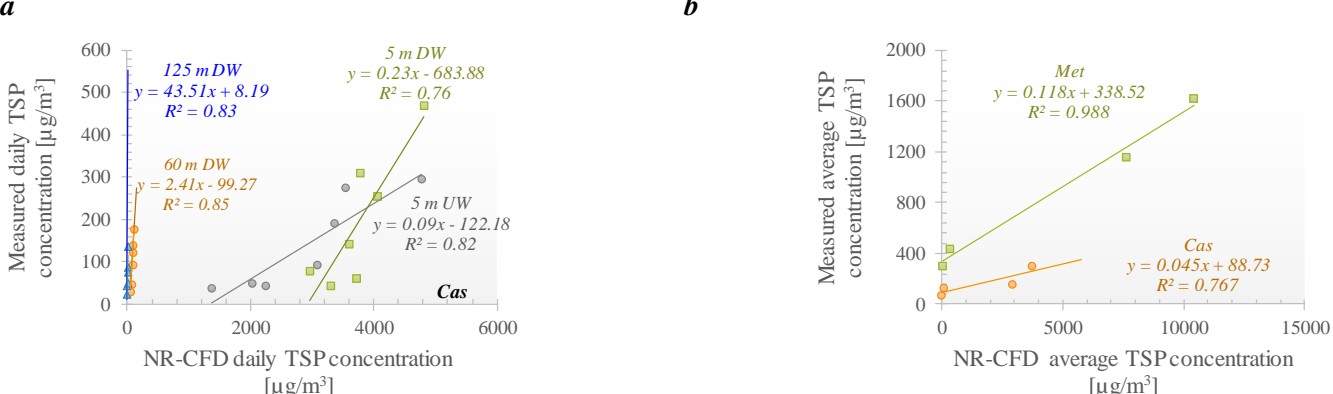

**Figure 5 Calibration of the NR-CFD model. Comparison of a) daily and b) long-term average TSP concentration obtained by the NR-CFD model against measured data as a function of distance to the road.**





**a**

**b**

**c**

**d**

**e**

**Figure 6 Non-dimensional TSP concentration at ground level as a function of non-dimensional distance to the road. Results obtained for different emission rates by a.) the AERMOD model with a given one-year meteorology data set, and b) the NR-CFD model with U\* =80. c.) Results obtained for different meteorological conditions by the AERMOD model, and d.) for different wind speeds by the NR-CFD model. e.) Comparison of the AERMOD and NR-CFD model results.**



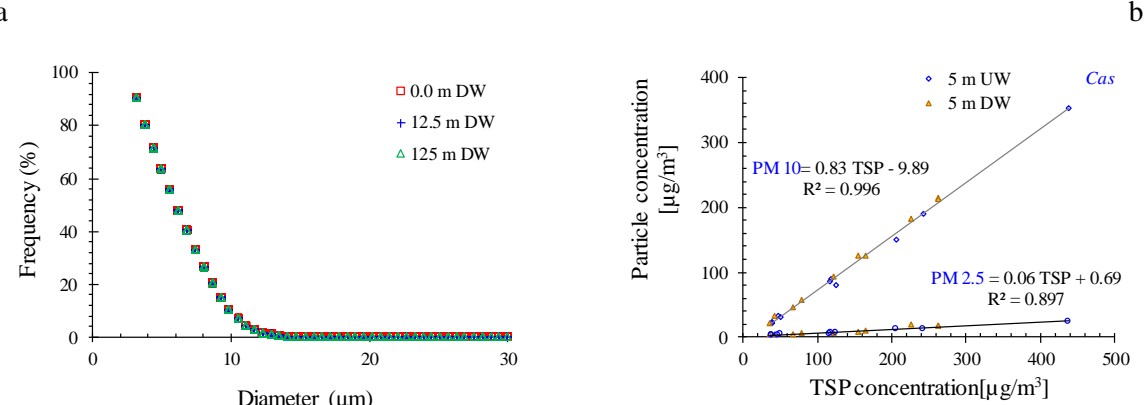

**Figure 7 a) Particle size distribution obtained via the NR-CFD model at different distance from the road Cas and b) comparison of experimental measurements of TSP against PM10 and PM2.5 obtained by the automatic stations located 5 m upwind and 5 m downwind road Cas.**

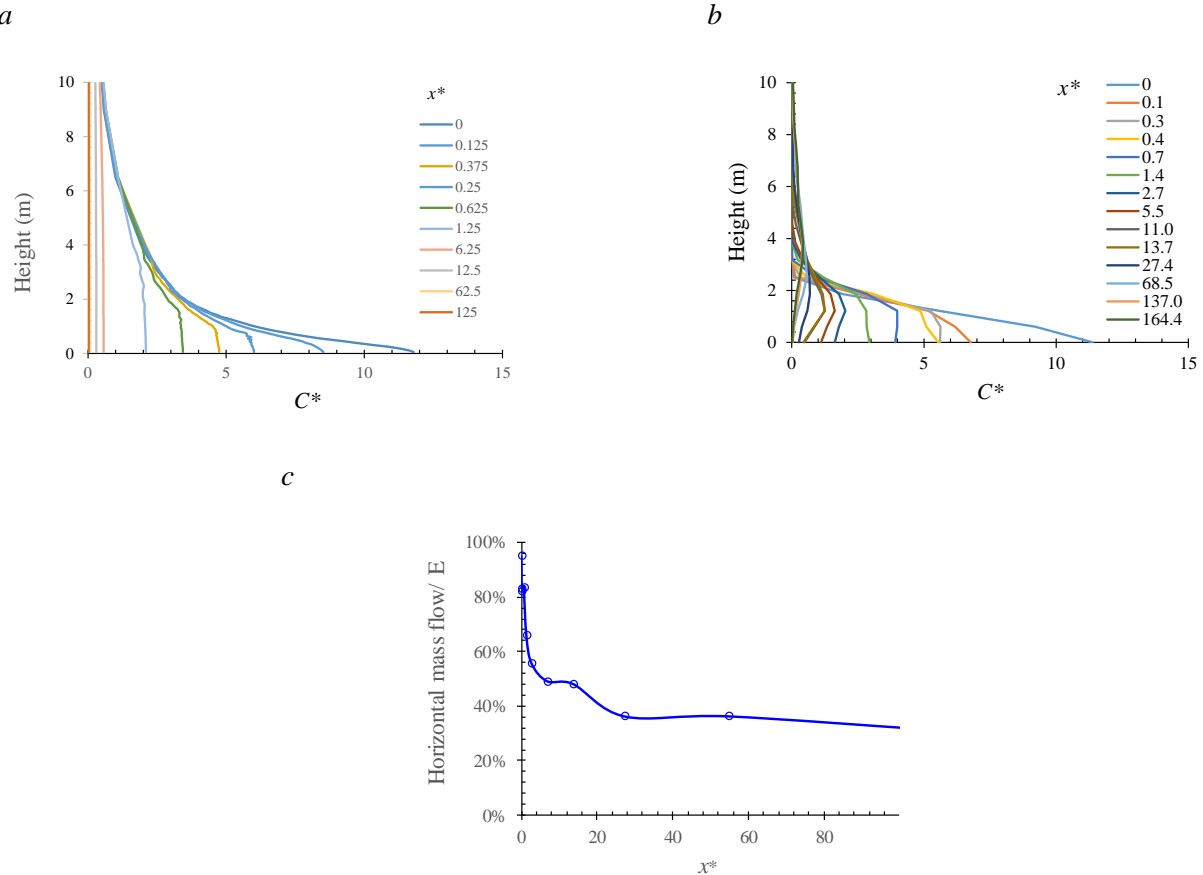

**Figure 8 Vertical TSP concentration profiles using the a) AERMOD, b) NR-CFD model, as function of distance to the road edge.**
5    **c.) Ratio of the horizontal mass flow and mass emission of particles as function of distance to the road edge.**





*a*                                                                                                                        *b*

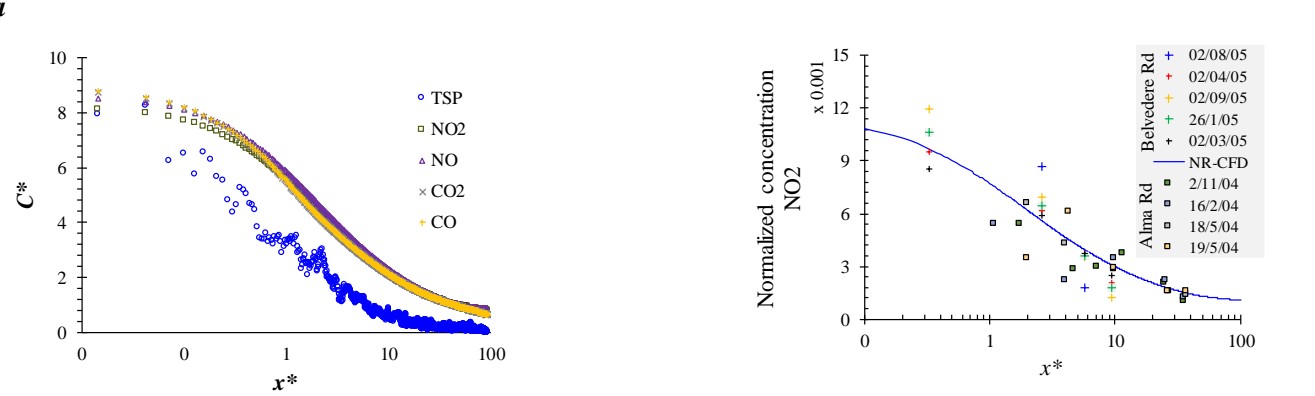

**Figure 9 a.) Results from the NR-CFD simulation of solid and gas-phase pollutant dispersion near roads for U\*=80. b.) Comparison of NO2 concentrations measured near two roads in UK (Chaney et al., 2011) and NR-CFD results, in terms of normalized concentrations as function of x\*.**



## Symbols and acronyms

| | Description | Units |
|---|---|---|
| $\alpha_1, \alpha_2$ | Beta function parameters | - |
| $a, b, k$ | Empirical constants | - |
| $C$ | TSP concentration at surface level | $\mu g/m^3$ |
| $C_{i,j}$ | TSP concentration for $i$ hour and distances $j$ from the road edge | $\mu g/m^3$ |
| $\overline{C_J}$ | Annual TSP concentration at distance $j$ from the road edge | $\mu g/m^3$ |
| $C^*$ | Non-dimensional TSP concentration | - |
| $d, \overline{d}$ | Particle diameter, Average particle diameter | $\mu m$ |
| $E$ | TSP mass emission rate per road area | $g /s\, m^2$ |
| $E_{fi}$ | Emission factor for vehicle of size $i$ in kg of TSP per vehicle and per km traveled | $kg/VKT$ |
| $F_d$ | Dispersion factor | - |
| $f_{PM10} - f_{PM2.5}$ | Constant fraction of the estimated concentration | - |
| $f_{k,q}$ | The frequency at which wind speed of intensity $k$ appears in the wind rose for direction $q$ | - |
| $g_z$ | Gravity | $m/s^2$ |
| $K$ | Von Karman universal constant | - |
| $L$ | Road width | $m$ |
| $m$ | Number of rainy days in the period with precipitation levels exceeding 0.254 mm | $days$ |
| $\mu$ | Fluid molecular viscosity | $kg/m\cdot s$ |
| $n$ | Spread parameter of the particle size distribution function | - |
| $n_d$ | Number of days in the period | $days$ |
| $N_i$ | Number of vehicle of size $i$ | - |
| $\eta_r$ | Efficiency of particulate matter emission control by rain | - |
| $\eta_s$ | Efficiency of particle emission control by water spraying | - |
| $p$ | Average daytime evaporation rate | $mm/h$ |
| $q$ | Irrigation application intensity | $L/m^2$ |
| $r$ | Average daily traffic | $Veh/h$ |
| $R_{c,i}$ | Mass emission ratio | |
| $\rho, \rho_p$ | Fluid and particle density | $kg/m^3$ |
| $s$ | Silt content of road surface material | - |
| $Sc$ | Schmidt number | |
| $t$ | Average time between spray applications | $h$ |
| $u^*$ | Friction velocity | $m/s$ |
| $u$ | Local fluid velocity in the x direction | $m/s$ |
| $U$ | Mean wind speed in the x direction | |
| $U^*$ | Non-dimensional speed ratio | - |
| $V_s$ | Settling speed | $m/s$ |
| $W$ | Average weight of the vehicles of size $j$ traveling in the road | $Tons$ |
| $w_{po}$ | Particle emission speed | $m/s$ |
| $x$ | Distance from the road edge | $m$ |
| $x_e$ | Equivalent distance from the road | |
| $x^*$ | Non dimensional distance to the road edge | |
| $Y_d$ | Rossin Rammler particle size distribution | - |
| $Z$ | Pre-exponential factor | - |
| $z$ | Height | $m$ |
| $z_o$ | Surface roughness | $m$ |
| $y+$ | Non dimensional wall distance | - |
| $CFD$ | Computational fluid dynamics | |
| $FB$ | Fractional bias | |
| $PM_{10}$ | Particulate matter with aerodynamic diameter < 10 $\mu m$ | |
| $PM_{2.5}$ | Particulate matter with aerodynamic diameter < 2.5 $\mu m$ | |
| $RMSE$ | Root mean square error | |
| $SEM-EDS$ | Scanning electron microscope coupled with energy-dispersive x-ray spectrometry | |
| $TSP$ | Particulate matter with aerodynamic diameter <30 $\mu m$ | |



| Description | Units |
|---|---|
| *XPS* | X-ray photoelectron spectroscopy technique | |