# Peer review of "Air pollution near arterial roads: An experimental and modelling study"

_Atmospheric Chemistry and Physics, 2017_

## Referee Comment (RC1) · Anonymous Referee #1 · 25 Oct 2017

In this paper, the authors present measurements and modeling of air pollutant dispersion downwind of an arterial road. Measurements of TSP, PM10 and PM2.5 concentrations were collected at several downwind locations, along with traffic and meteorological variables. The authors also performed modeling of the dispersion of particulate and gaseous pollutants in a near-road environment using CFD modeling tool, and the simulated results were compared with their measurements. The authors claim that a unique curve (beta function) can describe the near-road dispersion of various air pollutants. While the topic is important because vehicle emissions undergo a rapid transformation in a near-road setting, which influences the physiochemical nature of pollutants and has implications for health and environmental effects and relevant for

this journal, I thus do have some major concerns and comments about presentation, analysis, and discussions in this paper, which are outlined below:

General comment: The analysis and discussion in this paper are not comprehensive. Overall, this paper is not well written. The authors tried to do so many things in this article, but there is no good story. There are lots of sections and sub-sections where the reader cannot get the full picture. Discussions are limited in many cases, and many conclusions are not well substantiated by their analysis results. It seems like the authors are trying to include several aspects in this paper, but there is no comprehensive view. The QA/QC of data is not well documented. Specific comments are given in bellow.

Major comments: In this paper, the authors measured and modeled coarse (TSP, PM10, PM2.5) PM fractions downwind of two arterial roads. Many previous near-road studies have demonstrated that the traffic-emitted PM in a near-road setting is mostly dominated by ultrafine particles (< 100 nm), whereas coarse (TSP, PM10, PM2.5) PM is mainly dominated by regional/local background particles. The traffic-related pollutants (ultrafine particle, BC, NOx, CO, etc.) have strong near-road gradients, whereas near-road gradient of coarse PM is typically very mild (Karner et al., 2010). So one could have a strong concern that at which extent their measurements are relevant to traffic emissions? Although the authors performed their measurements at downwind of unpaved road, therefore, a large fraction of measured coarse PM are coming from the road dust. My concern is that how their measurements are relevant for a typical traffic emission/near-road perspective. For example, if someone wants to apply the knowledge from their paper in a typical near-road setting. Since the title of their paper says 'Air pollution near arterial roads'- thus, someone might expect the influence of traffic-related pollutants (combustion pollutants; ultrafine particle, BC, NOx, CO, etc.) at first, not that much about coarse PM. I think the author should have a strong justification on how their measurements fit in a context of typical traffic-emissions/near-road environment. If the coarse PM is critical for a traffic/near-road perspective under particular

environment, then the authors should reframe their paper, its title and analysis- center around coarse PM (since they only have coarse PM measurements) and that particular environment. As it is, to me, their measurements and analysis do not represent a typical near-road/traffic-related pollutants scenario.

Other specific comments: 1) Background vs. roadway impact: How did the authors separate re-suspended PM from unpaved road vs. traffic emitted larger particles? This is important if exploring the influence of traffic emission is a primary goal of their study?

2) Method section: there should be a clear description of what they measured, what instruments they used, how did they maintain QA/QC and data quality, instrument response time, data averaging time, sampling frequency, etc. These are very important given the near-road environments are very dynamic, in general. The detail on these can put in the supplementary. A table should be given summarizing all the important aspects related to instrumentations and data quality. There is no details about their sampling, variability, measurement uncertainty, etc. Did they measure continuously? How many sample they collected at different locations and for how many days? There is no real mention (Fig) about their measured data and its variability. Also, based on their 24-hr filter sample, how did they tell anything about traffic influence since traffic is very dynamic? With their 24 hour filter sample, they essentially do not have any temporal information. For example, the influence of meteorology (boundary layer ariation), traffic (diurnal traffic variation).

3) PM size distribution and composition: It is very confusing that they frequently generalized PM without mentioning any size information. What they measured is road dust (PM10 and TSP). Traffic emitted particles are dominated by smaller particles (a majority of combustion particle). What traffic-related info they might get based on filter SEM analysis of coarse PM? They reported that changes in particle size distribution are negligible within ∼1 km from the road edge, which is very confusing and miss-leading. First, their measurements are mostly road-dust, not traffic particles, so there should not be any significant gradient for that. In reality, the size distribution of traffic-emitted

particle in a near-road environment is highly dynamic and changes very rapidly within a few hundred meters from the roadway (Zhang and Wexler, 2004). Several complex microphysical processes dictate that changes, such as dilution, evaporation, condensation, coagulation, etc. Since they only measured TSP, which is not that traffic-related. Therefore, their results would not tell the true nature of the typical traffic-related particle.

4) Traffic data (Page 4): how did they measure traffic data? Details should be given about measurement technique, data averaging time and data quality. Also, it is important to have some information about fuel use scenario (diesel vs. gasoline use). The reported traffic flow rate (20-50 veh./hr) looks very unreasonable to me, especially for an arterial road.

5) P4: There are a bunch of equations, but there is no description of what are they and what is the meaning of different symbols. There is a list of symbols at the end, but it's good to have the description of symbol along with equation. Also, how did they get inputs for estimating EF, which is not clear to me? Clarification is needed.

6) They reported that the non-dimensional concentration of all gas phase pollutants exhibits a unique profile (Figure 9.a) that can be represented by a beta function with parameters. This is something over-weighted (more generalized) to me. Can the author model the concentration profile from different seasons using their unique function? I'd expect a substantial seasonality on near-road pollutant gradients. Can their unique function account the seasonality and different physicochemical transformation of different pollutants as well? Did they test it? Otherwise, this conclusion might be very misleading.

7) Vertical profile of PM distribution: Did they measure it? Can they evaluate their model results? TSP concentrations in an unpaved road would be highest at ground level that makes sense. But, for traffic emitted pollutants (e.g., ultrafine particles), it could be very different. They should be very careful while reporting different PM fraction. They should not generalize PM without any size information. This is very

confusing throughout the paper.

8) P10, L47: "we used the NR-CFD model to study differences in the dispersion of CO, CO2, NO2 and TSP"- Did they measure these gases? There is no description on that?

9) P5L1: "primary and secondary meteorological variables"- not sure what did they mean by primary and secondary met variable here? Which are primary and which are secondary?

10) "Particles exhibit a Rosin Rambler size distribution with average diameter of $\sim 7$ $\mu$m" – This is again very confusing. What did they mean by particles here? Particle mass or number size distribution? It seems PM mass. However, how relevant is this in context of traffic-emitted particles? I guess, this is only telling something about road-dust, not much about traffic-emitted PM. Clarification is needed.

Karner, A. A., Eisinger, D. S. and Niemeier, D. A.: Near-Roadway Air Quality: Synthesizing the Findings from Real-World Data, Environ. Sci. Technol., 44(14), 5334–5344, doi:10.1021/es100008x, 2010.

Zhang, K. M. and Wexler, A. S.: Evolution of particle number distribution near roadways—Part I: analysis of aerosol dynamics and its implications for engine emission measurement, Atmos. Environ., 38(38), 6643–6653, doi:10.1016/j.atmosenv.2004.06.043, 2004.

---

## Author Comment (AC1) · 21 Nov 2017

**José Ignacio Huertas Cardozo and Daniel Fernando Prato Sánchez**

daniel.pratto@gmail.com

For a more comprehensive understanding of the author's response, we invite the Referee to see it in the supplement material.

Air pollution near roads: An experimental and modelling study

Reply to reviewer 1 Nov 2017 Referee #1 Comments: General comment. The analysis and discussion in this paper are not comprehensive. Overall, this paper is not well written. The authors tried to do so many things in this article, but there is no good story. There are lots of sections and sub-sections where the reader cannot get the

full picture. Discussions are limited in many cases, and many conclusions are not well substantiated by their analysis results. It seems like the authors are trying to include several aspects in this paper, but there is no comprehensive view. The QA/QC of data is not well documented.

Reply: We thanks comments from our reviewer and appreciate his/her effort to provide comments to improve our manuscript and our work.

Before replaying this general comment, we clarify that we use the following terminology regarding particles. PM (Particulate matter): solid phase particles, regardless of their size, particle size distribution, morphology or chemical composition TSP (Total suspended particles or fine particles). Particles with aerodynamic diameter <30 um, PM10: particles with aerodynamic diameter <10 um, PM2.5: particles with aerodynamic diameter <2.5 um, UFP: particles with diameters in the range of 1-100 nm.

We oriented this manuscript towards the description of the temporal and spatial variations of traffic related pollutants near non- urban roadways by using a calibrated model which solve, via CFD, the equations that model the physics of dispersion of solid and gas phase species in a gas phase media under the varying conditions of the low troposphere (Pg 3, line 9). We highlight that:

We did not orient this manuscript towards the description of the CFD model because i.) it will make the manuscript too long, ii.) we believe that those contributions are not of the interest of the ACP audience. Then, we decided to include in this manuscript a brief description of the most relevant aspects of the model and its experimental validation. We also decided to fully describe the model in a companion paper, which is already under evaluation. (Pg 3, lines 10-13) We did not orient our manuscript towards the description of the spatial variations of traffic-related pollutants near roads based only on our experimental measurements because: The purpose of the experimental work was to validate our NR-CFD model and then, use the calibrated NR-CFD model to study the effect of the varying conditions of traffic on near-road pollutant concentration,

in terms of short and long-term concentrations, so that they can be used to assess human health impact. It will limit the validity of our conclusions to the specific conditions and timeframes under which we developed our experimental work (unpaved roads, 1 month of measurements, in a tropical area).

Despite the contributions related to the implementation of the NR-CFD model are being published elsewhere, this manuscript reports the most important contribution of our overall work (pg 3, line 9).

We systematically used our NR-CFD model to determine concentration profiles downwind in the horizontal and vertical direction as a function of meteorology parameters, emission rates and physical properties of the pollutant. We proposed a non-dimensional number for pollutant concentration, distance from the road and showed that all gas-phase pollutants exhibit the same profile, and all solid phase pollutants exhibit the same profile. We developed a methodology to include the temporal variations in the analysis and to provide integrated long-term averages of concentration downwind, which are useful to evaluate the human health impact of those pollutants. We measured near two unpaved roads, for long periods of time ($\sim$1 month), simultaneously, meteorological variables, traffic conditions, PM2.5, PM10 and TSP concentrations at 4 locations near the road, using instrumentation and protocols recommended by the USEPA and WMO.

The manuscript was modified to emphasize on this orientation and the scope of our work. We also improved the description of the experimental work and results obtained from those measurements.

Major comments. In this paper, the authors measured and modeled coarse (TSP, PM10, PM2.5) PM fractions downwind of two arterial roads. Many previous near-road studies have demonstrated that the traffic-emitted PM in a near-road setting is mostly dominated by ultrafine particles (< 100 nm), whereas coarse (TSP, PM10, PM2.5) PM is mainly dominated by regional/local background particles. The traffic-related pollu-

tants (ultrafine particle, BC, NOx, CO, etc.) have strong near-road gradients, whereas near-road gradient of coarse PM is typically very mild (Karner et al., 2010). So one could have a strong concern that at which extent their measurements are relevant to traffic emissions? Although the authors performed their measurements at downwind of unpaved road, therefore, a large fraction of measured coarse PM is coming from the road dust. My concern is that how their measurements are relevant for a typical traffic emission/near-road perspective. For example, if someone wants to apply the knowledge from their paper in a typical near-road setting. Since the title of their paper says 'Air pollution near arterial roads'- thus, someone might expect the influence of traffic-related pollutants (combustion pollutants; ultrafine particle, BC, NOx, CO, etc.) at first, not that much about coarse PM. I think the author should have a strong justification on how their measurements fit in a context of typical traffic-emissions/near-road environment. If the coarse PM is critical for a traffic/near-road perspective under particular environment, then the authors should reframe their paper, its title and analysis- center around coarse PM (since they only have coarse PM measurements) and that particular environment. As it is, to me, their measurements and analysis do not represent a typical near-road/traffic-related pollutants scenario.

Reply: ok. The manuscript was modified. We understood that the main concern of the reviewer is that our measurements and modelling work included fine particles (0.1 <d <30 um) but did not include UFP (d< 100 nm). Thus, our manuscript is not describing the dispersion of pollutants near a typical near-road environment, which our reviewer consider to be the one near paved roads with high traffic of diesel and gasoline-fueled vehicles.

We are looking for a larger scope than only paved roads and UFP. As stated previously, the focus of this manuscript is the description of the dispersion of pollutants (fine particles, UFP, non-reactive gases) near roads (paved and unpaved) by solving the differential equations that describe the physics of dispersion of gas and solid phase species in a gas phase media under the conditions of the low troposphere. Therefore:

We used the experimental work to calibrate this NR-CFD model for the case of fine particles We used the calibrated NR-CFD model and the experimental measurements to describe the spatial and variations of fine particles near unpaved roads. We extrapolated the use of our calibrated NR-CFD model to study the dispersion of gases near paved and unpaved roads. We found that results agree with reported experimental results (Chaney, Cryer, Nicholl, and Seakins, 2011). We highlight that pollutants near roads, besides tailpipe pollutants and pollutants present in the background, includes pollutants resulting from the interaction tire-road (see replay to comment No. 1). Thus, re-suspended road particles are also traffic-related pollutants and they are also present in paved roads, independently if the vehicles moving on the road surface are powered by gasoline/diesel engines o electric motors.

Then: Attending reviewers concern, we are including a new section, where we again extrapolated the use of our NR-CFD model to the case of UFP for the case of the paved roads with high traffic of gasoline or diesel-fueled vehicles. We also compared to experimental measurements reported in the literature and found good agreement. See the description of UFP dispersion at the end of this document. We are highlighting in the manuscript that the physics of dispersion of UFP is different that of physics of dispersion of fine and coarse particles.

Specific comments

Specific Comment 1. Background vs. roadway impact: How did the authors separate re-suspended PM from unpaved road vs. traffic emitted larger particles? This is important if exploring the influence of traffic emission is a primary goal of their study?

Reply: We understood that reviewer refers to traffic emitted particles as those particles emitted from the vehicle exhaust tube and traffic emitted larger particles as background particles.

We clarify that near-road pollutants include: Background pollutants, which are those pollutants originated from the surroundings, exhibit a constant concentration, and depend on the specific place of study. Thus, they become relevant when we are interpreting experimental data or when we are considering interactions between background pollutants and the ones originated from the source under study. Traffic-related pollutants: Pollutants originated from the circulation of vehicles on the road. These pollutants can be classified according to their origin in: Tailpipe emissions: These pollutants are originated from the combustion processes of engine-powered vehicles. They include ultrafine particles (UFP), CO, CO2, NOx, SOx, and unburned hydrocarbons, among which VOCs are of high interest. Those pollutants are of high interest, especially for the case due to their human impact. These pollutants are not present in the case of electric-powered vehicles. Tire-road interactions: Emission of coarse (d >30 ïA■m), fine (0.1 <d <30 ïA■m) and ultrafine (d< 100 nm) particles result from the tire-road interaction due to tire wear, breaks wear, and resuspension of road particles. This source of particles is always present in both paved and unpaved roads.

Then: Traffic-related emissions include both: tailpipe emissions and particles originated from the tire-road interactions. In the study of near-road pollution, we should include both types of particles. In the case of unpaved roads, particle emissions due to resuspension are at least four orders of magnitude higher than tailpipe emissions. We identified that particles trapped in the high-vol filters came from resuspension and we could not identify tailpipe particles in the SEM analysis. This was due to the fact that tail-pipe particles had a negligible concentration compared to the ones originated from resuspension. We did not intent to measure UFP in our experimental work. Background particles are different from re-suspended particles due to the interaction tire-road. In our experimental work, we selected two regions with negligible particle background concentration. We selected two regions fully covered with grass and no houses or any source of particles in a circle of at least 1 km of radius.

In conclusion, we did not need to separate tailpipe particles from re-suspended particles to accomplish our objective of calibrate our NR-CFD model.

Specific Comment 2. Method section: there should be a clear description of what they

measured, what instruments they used, how did they maintain QA/QC and data quality, instrument response time, data averaging time, sampling frequency, etc. These are very important given the near-road environments are very dynamic, in general. The detail on these can put in the supplementary. A table should be given summarizing all the important aspects related to instrumentations and data quality. There is no details about their sampling, variability, measurement uncertainty, etc. Did they measure continuously? How many sample they collected at different locations and for how many days? There is no real mention (Fig) about their measured data and its variability. Also, based on their 24-hr filter sample, how did they tell anything about traffic influence since traffic is very dynamic? With their 24 hour filter sample, they essentially do not have any temporal information. For example, the influence of meteorology (boundary layer variation), traffic (diurnal traffic variation).

Reply: ok. We are including additional details of the experimental measurements and data analysis. We are also including the measured data as supplementary material.

As our driver is the assessment of near-road air pollution on human health, we are interested in short (1, 8, 24 hr) and long (1 year) term averages of pollutant concentration downwind from the road to contrast them with air quality standards. The air quality standards are the max values of exposure (concentration during a given period of time) below which studies have shown to be safe for human health.

From the experimental perspective, the USEPA have stablished the recommended practices to determine pollutants concentration, which includes instrumentation technical characteristics and measurement protocols. Those protocols stablish the duration of individual measurements. For example, the USEPA have stablishes the determination of PM10 concentration through weight differences of filters exposed during 24 hr to a constant volumetric flow. We followed those protocols and determined PM10 and TSP concentration simultaneously in 4 points near the road.

The USEPA have also incorporated within its recommended practices the use nephelometer (light attenuation technique) for the determination of PM10 and PM2.5. This technique allows to have 1 min measurements, but for air quality assessment those measurements should be averaged for periods of 24 hrs. We incorporated two instruments that use this technique in our measurements. We measured every 10 min and reported 24 hrs-averaged values. (pg 5 line 10).

For the case of the meteorological variable, the World Meteorological Organization (MWO) have established the instruments technical characteristics and measurements protocols recommended for their determination (World Meteorological Organization, 2008). We followed those recommended practices. Measurements were reported as 1 hr-averaged values.

There is no real mention (Fig) about their measured data and its variability. Also, based on their 24-hr filter sample, how did they tell anything about traffic influence since traffic is very dynamic? With their 24 hour filter sample, they essentially do not have any temporal information. For example, the influence of meteorology (boundary layer variation), traffic (diurnal traffic variation).

Attending reviewer suggestion, we included in the manuscript figures describing the temporal variations of the measured data.

We did not oriented this manuscript towards the description of PM10 and TSP dispersion based only on our experimental measurements because it will limit the validity of our conclusions to the specific conditions and timeframes under which we developed our experimental work. We neither intended to correlate TSP and PM10 concentrations with traffic and meteorological conditions because we chose to study the influence of those variables on pollutant concentration through the physics of dispersion included in the NR-CFD model.

Again, the purpose of the experimental work was to validate our NR-CFD model. Then, we used the calibrated NR-CFD model to study the effect of the varying conditions of traffic on near-road pollutant concentration, in terms of short and long-term concentra-

Interactive
comment

tions, so that they can be used to assess human health impact. Using 1 hour values for traffic conditions and meteorological variables, we modeled the dispersion of fine particles every hour assuming that steady state conditions prevail within each 1-hour time interval. Then we averaged those results for periods of 24 hrs and compared them with experimental measurements. We also compared modeled and experimental results in terms of 1 month averaged values. As stated in the manuscript we found high correlations among them indicating the NR-CFD model and our approach of 1 hr modelling predicts well short (24 hrs) and long-term (>1 month) concentrations.

Specific Comment 3. PM size distribution and composition: It is very confusing that they frequently generalized PM without mentioning any size information. What they measured is road dust (PM10 and TSP). Traffic emitted particles are dominated by smaller particles (a majority of combustion particle). What traffic-related info they might get based on filter SEM analysis of coarse PM? They reported that changes in particle size distribution are negligible within _1 km from the road edge, which is very confusing and miss-leading. First, their measurements are mostly road-dust, not traffic particles, so there should not be any significant gradient for that. In reality, the size distribution of traffic-emitted particle in a near-road environment is highly dynamic and changes very rapidly within a few hundred meters from the roadway (Zhang and Wexler, 2004). Several complex microphysical processes dictate that changes, such as dilution, evaporation, condensation, coagulation, etc. Since they only measured TSP, which is not that traffic-related. Therefore, their results would not tell the true nature of the typical traffic-related particle.

Reply: ok,the manuscript was modified.

It is very confusing that they frequently generalized PM without mentioning any size information We use the following terminology regarding particles. PM (Particulate matter): solid phase particles, regardless of their size, particle size distribution, morphology or chemical composition TSP (Total suspended particles or fine particles). Particles with aerodynamic diameter <30 ïA∎m, PM10: particles with aerodynamic diameter

<10 ïA■m, PM2.5: particles with aerodynamic diameter <2.5 ïA■m, UFP: particles with diameters in the range of 1-100 nm

What they measured is road dust (PM10 and TSP). Yes. We also measured PM2.5. (Pg 5, line 5). As described previously, they are traffic-related pollutants.

Traffic emitted particles are dominated by smaller particles (a majority of combustion particle). In the general case, particles emitted from roads include: i.) Particles emitted from the vehicle tailpipe (exhaust emissions) ii.) Particle emitted due to wear and tear of vehicle parts such as brake, tyre and clutch iii.) and re-suspension of particles (non-exhaust emissions) (Pant and Harrison, 2013). It has been shown that even with zero tail-pipe emissions, traffic will continue to contribute to fine and ultrafine particles through non-exhaust emissions (Briefs & Environmental, n.d.)(Dahl et al., 2006; Kumar et al., 2013) and it is estimated that nearly 90% of the total emissions from road traffic will come from non-exhaust sources by the end of the decade (Rexeis and Hausberger, 2009). Non-exhaust emissions, are becoming more important now, and further research is anticipated in this field in the coming years (Pant and Harrison, 2013).

What traffic-related info they might get based on filter SEM analysis of coarse PM We did not intent to determine UFP concentration during our experimental work. As stated before, traffic related emissions include both: tailpipe emission and re-suspended particles. SEM analysis confirmed that particle trapped in the filters came from the resuspension of particles of the same chemical composition that the road material. Based on SEM analysis we also obtained particle size distribution. SEM analysis did not show the presence of particles originated from combustion processes, which was expected because particle emissions due to resuspension is 4 orders of magnitude higher than tailpipe particle emissions.

They reported that changes in particle size distribution are negligible within _1 km from the road edge, which is very confusing and miss-leading. In reality, the size distribution of traffic-emitted particle in a near-road environment is highly dynamic and changes

very rapidly within a few hundred meters from the roadway (Zhang and Wexler, 2004). Several complex microphysical processes dictate that changes, such as dilution, evaporation, condensation, coagulation, etc. Since they only measured TSP, which is not that traffic-related. Therefore, their results would not tell the true nature of the typical traffic-related particle. Experimentally and trough simulation we obtained that particle size distribution for the case of fine particles (1<d<30 um) remains essentially constant within the first km from the road. As stated on the manuscript (pg 9, line 10), several other author have reached the same conclusion (Zhu et al., 2011). We modified the manuscript and clarified that this conclusion may not be true for the case of UFP.

We added a new paragraph discussing the dispersion of UFP and the results obtained by our NR-CFD model. See the end of this document.

Specific Comment 4. Traffic data (Page 4): how did they measure traffic data? Details should be given about measurement technique, data averaging time and data quality. Also, it is important to have some information about fuel use scenario (diesel vs. gasoline use). The reported traffic flow rate (20-50 veh./hr) looks very unreasonable to me, specially for an arterial road.

Reply: ok, the manuscript was modified. We included the additional information that reviewer suggested about the measurement campaign and data treatment related to traffic data.

Yes, our traffic flow is too low for an arterial road. We did not state that we performed our experimental work near an arterial road. We did it near a local road and specifically near two unpaved roads (pg 3, lines 30). The purpose of our experimental work was to validate the NR_CFD model. We selected unpaved roads for this purpose because the procedures and instrumentation used to measure PM2.5, PM10 and TSP concentration are well stablished. Besides that, near unpaved roads PM2.5, PM10 and TSP concentrations are much higher that the uncertainties involved in the measurement procedures.

We also modified the title of our manuscript.

Specific Comment 5. P4: There are a bunch of equations, but there is no description of what are they and what is the meaning of different symbols. There is a list of symbols at the end, but it's good to have the description of symbol along with equation. Also, how did they get inputs for estimating EF, which is not clear to me? Clarification is needed.

Reply: ok, the manuscript was modified. We included a description of symbols along with the equations. We also clarified on the data used to estimate EF.

The following equations estimate the mass of fine particles emitted from paved and unpaved roads, E=$(\sum N\_jE\_fij)/3600L(1-\eta$\_r$)$(1-$\eta$\_rn) (1)

where E TSP mass emission rate per road area g /s m2 E_fi Emission factor for vehicle of size i in kg of TSP per vehicle and per km traveled kg/VKT L Road width m $\eta$\_r Efficiency of particulate matter emission control by rain - $\eta$\_s Efficiency of particle emission control by water spraying -

The USEPA recommends to use the following Efi for the case of TSP and PM10 emissions from paved and unpaved roads.

Table 1.

Please see it in the supplementary material

Specific Comment 6. They reported that the non-dimensional concentration of all gas phase pollutants exhibits a unique profile (Figure 9.a) that can be represented by a beta function with parameters. This is something over-weighted (more generalized) to me. Can the author model the concentration profile from different seasons using their unique function? I'd expect a substantial seasonality on near-road pollutant gradients. Can their unique function account the seasonality and different physicochemical transformation of different pollutants as well? Did they test it? Otherwise, this conclusion might be very misleading.

Reply: These are the major assumptions of our NR-CFD model: At every hour, dispersion occurs under pseudo steady state conditions. Dispersion happens on a flat terrain with no obstacles to the dispersion of pollutants and the road is the only source of pollutants Temperature remains constant within the computational domain ($\sim$ 50 m of height) No chemical reactions (or phase changes) occur.

We confirm that under these circumstances, the dispersion of all gas phase pollutants result in a unique profile of concentration vs distance to the road edge when expressed in terms of the non-dimensional numbers described in the manuscript. We also confirm that this profile can be described by a beta function with the parameters described in the manuscript.

We did test this observation under very diverse conditions of gas dispersion such as wind speed, emission rates and gas properties. We spent lot of effort looking for the set of appropriate variables that make the non-dimensional concentration profile unique.

Our results do account for seasonality. Even though the non-dimensional concentration vs distance profile remain the same, the actual (dimensional) gas concentration vs. distance changes with seasons. The main effect is due to temperature changes. Temperature changes diffusivity, density and viscosity of pollutants and gas-phase media. Even though atmospheric conditions change, the physics of dispersion remain the same.

Specific Comment 7. Vertical profile of PM distribution: Did they measure it? Can they evaluate their model results? TSP concentrations in an unpaved road would be highest at ground level that makes sense. But, for traffic emitted pollutants (e.g., ultrafine particles), it could be very different. They should be very careful while reporting different PM fraction. They should not generalize PM without any size information. This is very confusing throughout the paper.

Reply: Ok, we modified the manuscript to clarify the type of particles that we are referring to. We did not measure any vertical particle concentration profile. We compared

qualitatively our results on TSP vertical concentration profile with experimental results reported in the literature (Yuan, Ng, and Norford, 2014;Shen, Cui, and Zhang, 2017; Kwak, Baik, Ryu, and Lee, 2015). We found that they exhibit the same profile.

See the end of this document for results on the vertical profile of UFP obtained by our NR-CFD model.

Specific Comment 8. P10, L47: "we used the NR-CFD model to study differences in the dispersion of CO, CO2, NO2 and TSP"- Did they measure these gases? There is no description on that?

Reply: Ok We did not measured these gases. In P11, L7, we stated "Aiming to validate these results, we looked in the literature for experimental data. Several works have reported measurements of gas phase pollutants concentration near roads. However, none of them reports simultaneous measurements of mass emissions, meteorological conditions and pollutants concentration that could be used to validate quantitatively the NR-CFD model. As an approximation, we compared, qualitatively, NR-CFD results to values of NO2 measured near roads with high traffic of heavy-duty vehicles in UK (Chaney et al., 2011). Each set of simultaneous NO2 measurements were normalized in the way that the area under the concentration vs distance to the road edge curve were equals to one. Figure 9.b shows that numerical and experimental results are similar. Non-dimensional concentration differs from normalized concentration in their area under the curve but their shapes are similar.

Specific Comment 9. P5L1: "primary and secondary meteorological variables"- not sure what did they mean by primary and secondary met variable here? Which are primary and which are secondary?

Reply: Ok. Manuscript was modified. We removed "and secondary" from the manuscript.

Specific Comment 10. "Particles exhibit a Rosin Rambler size distribution with average

diameter of _ 7 _m" – This is again very confusing. What did they mean by particles here? Particle mass or number size distribution? It seems PM mass. However, how relevant is this in context of traffic-emitted particles? I guess, this is only telling something about road dust, not much about traffic-emitted PM. Clarification is needed.

Reply: Manuscript was modified. We referred to particle number size distribution. We obtained it counting particles observed in SEM photographs.

References

Briefs, S., & Environmental, I. N. (n.d.). Parmod Kumar Surender Kumar Laxmi Joshi. Chaney, A. M., Cryer, D. J., Nicholl, E. J., & Seakins, P. W. (2011). NO and NO 2 interconversion downwind of two different line sources in suburban environments. Atmospheric Environment, 45(32), 5863–5871. https://doi.org/10.1016/j.atmosenv.2011.06.070 Kwak, K. H., Baik, J. J., Ryu, Y. H., & Lee, S. H. (2015). Urban air quality simulation in a high-rise building area using a CFD model coupled with mesoscale meteorological and chemistry-transport models. Atmospheric Environment, 100. https://doi.org/10.1016/j.atmosenv.2014.10.059 Pant, P., & Harrison, R. M. (2013). Estimation of the contribution of road traffic emissions to particulate matter concentrations from field measurements: A review. Atmospheric Environment, 77, 78–97. https://doi.org/10.1016/j.atmosenv.2013.04.028 Review Rexeis, M., & Hausberger, S. (2009). Trend of vehicle emission levels until 2020 - Prognosis based on current vehicle measurements and future emission legislation. Atmospheric Environment, 43(31), 4689–4698. https://doi.org/10.1016/j.atmosenv.2008.09.034 Shen, Z., Cui, G., & Zhang, Z. (2017). Turbulent dispersion of pollutants in urban-type canopies under stable stratification conditions. Atmospheric Environment, 156, 1–14. https://doi.org/10.1016/j.atmosenv.2017.02.017 US EPA. (2006). AP 42, Fifth Edition, Volume I Chapter 13: Miscellaneous Sources. Section 13.2.2. US EPA. (2011). AP 42, Fifth Edition, Volume I Chapter 13: Miscellaneous Sources. Section 13.2.1. Yuan, C., Ng, E., & Norford, L. K. (2014). Improving air quality in high-density cities by understanding the relationship between air pollutant

dispersion and urban morphologies. Building and Environment, 71(2), 245–258. https://doi.org/10.1016/j.buildenv.2013.10.008 Zhu, D., Kuhns, H. D., Gillies, J. A., Etyemezian, V., Gertler, A. W., & Brown, S. (2011). Inferring deposition velocities from changes in aerosol size distributions downwind of a roadway, 45, 957–966. https://doi.org/10.1016/j.atmosenv.2010.11.004

Best Regards Dr. José Ignacio Huertas School of Engineering and Science-EIC Automotive engineering research center- CIMA (http://cima.tol.itesm.mx/) Tecnológico de Monterrey (http://www.itesm.edu/) Phone: (52) 81 8358 2000 Ext 5293

Please also note the supplement to this comment:
https://www.atmos-chem-phys-discuss.net/acp-2017-753/acp-2017-753-AC1-supplement.pdf

---

## Referee Comment (RC2) · A.R. MacKenzie (Referee) · 11 Dec 2017

General remarks

This is an experimental and modelling study of pollutant dispersion from unpaved arterial roads. As written, and as described by the other reviewer, the manuscript is in danger of being misunderstood, and a careful re-write will be required before it can be considered for publication in ACP.

Specific comments

Title and abstract: should prefigure the main work of the paper more accurately. Please

include the word "unpaved" before "arterial roads".

Abstract. Please make it clear that the particle measurements in the study are 24-hour average mass concentrations. I don't see how emission mass rates were measured – please delete.

Abstract, Line 12. Please state whether these "plots of pollutants concentration" are measured or experimental. If they are experimental, please repeat the temporal averaging time.

P2, Line 20. You say "an important number of works" but only cite one work – please amend to make consistent.

P2, Line 30. The diffusivity differences between CO and NOx are trivial with respect to dispersion under turbulent mixing at a roadside. Also, CO and NOx disperse in the same plume, having the same density, not in plumes of different density, so I don't think this sentence is helpful. I suggest you delete.

P2, Line 32. Here, and thoughout, you must make it clear which size fraction you are discussing.

P2, Line 34. Gaussian models are not heuristic, see Seinfeld and Pandis, Atmospheric Physics and Chemistry, 2nd ed., Ch. 18.

P2, Line 37. This paragraph does not add anything to the manuscript and could be read as a criticism of the understanding of fluid dynamics of previous pioneers in the field. I suggest it is deleted.

P2, Line 44ff. There are many more studies of computational fluid dynamics for urban and rural roads than cited here. You should distinguish RANS approaches from large-eddy simulations and provide more citations. Your primary data to evaluate the model are 24-hour averages; please provide a discussion of why "state-of-the-art" CFD is the best method to interpret such long-time averages.

P3, Line 7. If the point of the previous paragraph was to introduce the idea of working with a commercial CFD package, then that package should be named here.

P3, line 16. Please re-write this bullet to state what the model is (not just "state of the art") and what it does (something more useful than to resolve a known issue with the Gaussian solutions).

P3, Line 21. Define symbols or direct reader to a list of symbols.

P3, line 24. Disambiguate TPS and TSP – are these the same or something different? Please provide horizontal scale bounds on the statement about "constant fraction" because as written it appears to break the laws of physics (or those laws conspire to match precisely the different loss processes affecting PM2.5 and TSP).

P3, Line 25. Please provide the reader with some idea of the threshold applied that allows an impact area to be defined.

Figure 1. Horizontal scales are given but not a vertical scale. The caption should draw the reader's attention to the non-uniform length scale.

P3, Line 29. Strictly, vegetation can be a source of primary (mostly coarse) aerosol particles composed of pollen, spores, or plant fragments. This sentence should be written more carefully.

P3, Line30ff. These sentences are rationale for the study and should be in the Introduction. When re-written in the correct place, this paragraph should carefully distinguish between sources which dominate the mass size distribution of roadside aerosol, and those which dominate the number size distribution.

P3, Line 38. Please state at which positions relative to the road which measurements were made, at which temporal resolution, and with which measuring equipment.

P4, Line 4. Please state that you will discuss model calibration in a later section.

P4, Line 28. Explain to the reader what changes to get emissions for TSP and PM10

from equations 1-4.

P5, Line 1. Please provide the temporal resolution of met data. Delete "primary and secondary".

P5, Line 17. Report briefly the definition of diameter derived from the microscopy (e.g., equivalent area, longest axis, etc).

P5, Line 17. Figure 2d reports apparently size in mm, not micrometres.

P5, line 18. Rosin-Rambler (abstract) or Rosin-Rammler?

P5, Line 35. Are these two different references? If so, they should be disambiguated.

P6, section 4.2. Whether an accompanying model description paper is available or not, sufficient detail of the modelling approach should be given to allow the reader to understand the model set-up and experiments. Please describe the model discretisation and turbulent closure as a minimum. Please provide some justification for the size of domain and for the boundary conditions chosen. Please provide details of model spin up to steady state. Please explain how 1 hour steady state models are to be compared to 24-hour average measurements. Please describe what microphysics, if any, is included in the model.

P6, Line 34ff. I don't believe that readers will accept you can model all kinds of vehicle-induced aerosol with a single 'quartz' model tracer following a Rosin-Rammler size distribution. It would be much more persuasive to stick to modelling the suspended silt that makes up the vast majority of the mass concentration in the hi-vol samples.

P7, Line 9. Finally, we learn that the model is FLUENT, set-up with a variety of standard settings. Please completely re-order the description of the modelling to start with the name of the commercial modelling system and describe the important set-up parameters as asked for above.

P8, Line 1. The calibration procedure is not clear. Calibration implies that some parts

of the measured data were used to refine model parameters and then the calibrated model used to simulate a different part of the data. Please explain.

Figure 5b. How long are the long-term averages? Please make all captions self-explanatory.

P8, Line 11. If the measurements are averages please also plot standard deviations (or, better, plot medians and quartiles).

P8, Line 16. Please report numbers using standard scientific notation. RMSE should have associated units.

P8, Line 26, It is a basic property of the Gaussian plume model that downwind concentrations are proportional to the emission rate, so Figure 6a is not needed. Figure 6b is more interesting, but only if some description of the model is provided that would account for non-linear behaviour with emission rate. Since the concentration further downwind is exactly proportional to emission rate for the CFD simulation, it is more pertinent to ask what is causing the spread near the source.

Figure 6b and 6d. I am not sure how "zero" can appear on a logarithmic scale. This will be confusing, especially for junior scientists, and should be removed.

P8, Line 30ff. It is, again, a standard result from Gaussian plume models that the concentration at a point varies inversely with wind speed. This para therefore shows what a good job Gaussian plume modelling does of capturing the time-average concentration profile downwind of a source, which has been demonstrated many times before. Again, the behaviour of the CFD for $x^* < 1$ is more interesting.

P9. If deposition is negligible and coagulation and condensational growth are not applicable/accounted for, then the ratios of size fractions are bound to remain constant.

P10, Line 10. It is not intuitive to expect a Gaussian vertical distribution from AERMOD. This would be the case for a chimney but not for a ground source.

P10, Line 46ff. Please explain why the model results for gas phase tracers are very much smoother than those for TSP.

P11, Line 23. This material should be much earlier on when the concept of 'area affected' is introduced. It is important to state what averaging time is used in the air quality standard you are using, and to compare similar modelled and measured averaging times.

P12. The conclusions should be re-written in light of the revisions suggested by the referees.

References. If Huertas and Prato is "in press" please provide the journal name.

Throughout: please could the font size and line spacing be made consistent.

---

## Author Comment (AC2) · 24 Jan 2018

**José Ignacio Huertas Cardozo and Daniel Fernando Prato Sánchez**

daniel.pratto@gmail.com

For a more comprehensive understanding of the author's response, we invite the Referee to see it in the supplement material.

Air pollution near unpaved roads: An experimental and modelling study

Reply to reviewer 2 Jan 2018

General remarks. This is an experimental and modelling study of pollutant dispersion from unpaved arterial roads. As written, and as described by the other reviewer, the manuscript is in danger of being misunderstood, and a careful re-write will be required

before it can be considered for publication in ACP. Reply: We thanks comments from our reviewer and appreciate his effort providing comments to improve our manuscript and our work.

Specifics comments. Specific Comment 1.Title and abstract: should prefigure the main work of the paper more accurately. Please include the word "unpaved" before "arterial roads". Reply: Ok. Manuscript was modified.

Specific Comment 2. Abstract. Please make it clear that the particle measurements in the study are 24-hour average mass concentrations. I don't see how emission mass rates were measured – please delete. Reply: Ok. Manuscript was modified

Specific Comment 3. Abstract, Line 12. Please state whether these "plots of pollutants concentration" are measured or experimental. If they are experimental, please repeat the temporal averaging time. Reply: Ok. Manuscript was modified

Specific Comment 4. P2, Line 20. You say "an important number of works" but only cite one work – please amend to make consistent. Reply: Ok. Manuscript was modified

Specific Comment 5. P2, Line 30. The diffusivity differences between CO and NOx are trivial with respect to dispersion under turbulent mixing at a roadside. Also, CO and NOx disperse in the same plume, having the same density, not in plumes of different density, so I don't think this sentence is helpful. I suggest you delete. Reply: Ok. Manuscript was modified We did not delete the whole sentence because it is relevant. We are emphasizing that, quantitatively, the dispersion of particles is not exactly the same as the dispersion of any gas phase pollutant.

Specific Comment 6. P2, Line 32. Here, and thoughout, you must make it clear which size fraction you are discussing. Reply: Ok. Manuscript was modified

Specific Comment 7. P2, Line 34. Gaussian models are not heuristic, see Seinfeld and Pandis, Atmospheric Physics and Chemistry, 2nd ed., Ch. 18. Reply: Ok. Manuscript was modified

Specific Comment 8. P2, Line 37. This paragraph does not add anything to the manuscript and could be read as a criticism of the understanding of fluid dynamics of previous pioneers in the field. I suggest it is deleted. Reply: Ok. Most of the paragraph was deleted.

Specific Comment 9. P2, Line 44ff. There are many more studies of computational fluid dynamics for urban and rural roads than cited here. You should distinguish RANS approaches from large eddy simulations and provide more citations. Your primary data to evaluate the model are 24-hour averages; please provide a discussion of why "state-of-the-art" CFD is the best method to interpret such long-time averages. Reply: Ok. Manuscript was modified

Specific Comment 10. P3, Line 7. If the point of the previous paragraph was to introduce the idea of working with a commercial CFD package, then that package should be named here. Reply: Ok. Manuscript was modified. We used Fluent v17 from ANSYS.

Specific Comment 11. P3, line 16. Please re-write this bullet to state what the model is (not just "state of the art") and what it does (something more useful than to resolve a known issue with the Gaussian solutions). Reply: Ok. Manuscript was modified

Specific Comment 12. P3, Line 21. Define symbols or direct reader to a list of symbols. Reply: Ok. Manuscript was modified

Specific Comment 13. P3, line 24. Disambiguate TPS and TSP – are these the same or something different? Please provide horizontal scale bounds on the statement about "constant fraction" because as written it appears to break the laws of physics (or those laws conspire to match precisely the different loss processes affecting PM2.5 and TSP). Reply: Ok. Manuscript was modified It should be TSP (total suspended particles).

Specific Comment 14. P3, Line 25. Please provide the reader with some idea of the threshold applied that allows an impact area to be defined. Reply: Ok. Manuscript

was modified. The environmental impact area generated by the use of the roads was defined as the area at both sides of the road where short or long-term average concentrations exceeds national air quality standards for any of the pollutants under consideration. We used thresholds of 100 and 300 ug/m3 for 24 h and annual averages of TSP concentrations, respectively, and thresholds of 50 and 100 ug/m3 for 24 h and annual averages of PM10 concentrations, respectively.

Specific Comment 15. Figure 1. Horizontal scales are given but not a vertical scale. The caption should draw the reader's attention to the non-uniform length scale. Reply: Ok, Figure 1 was modified

Specific Comment 16. P3, Line 29. Strictly, vegetation can be a source of primary (mostly coarse) aerosol particles composed of pollen, spores, or plant fragments. This sentence should be written more carefully. Reply: Ok. Manuscript was modified We selected a region in which the unique particulate matter (TSP, PM10 and PM2.5) emission source was the road. We selected areas covered with pastures and assumed that on these areas the emission of primary aerosol particles such as pollen, spores or plant fragments were negligible.

Specific Comment 17. P3, Line30ff. These sentences are rationale for the study and should be in the Introduction. When re-written in the correct place, this paragraph should carefully distinguish between sources which dominate the mass size distribution of roadside aerosol, and those which dominate the number size distribution. Reply: Ok. Manuscript was modified

Specific Comment 18. P3, Line 38. Please state at which positions relative to the road which measurements were made, at which temporal resolution, and with which measuring equipment. Reply: Ok. Manuscript was modified That information is provided later in the manuscript. We added the sentence: Section 2.1 will describe the experimental work conducted in this study.

Specific Comment 19. P4, Line 4. Please state that you will discuss model calibration

in a later section. Reply: Ok. Manuscript was modified

Specific Comment 20. P4, Line 28. Explain to the reader what changes to get emissions for TSP and PM10 from equations 1-4. Reply: Ok. Manuscript was modified. The emission factor Ef changes for TSP and PM10.

Specific Comment 21. P5, Line 1. Please provide the temporal resolution of met data. Delete "primary and secondary". Reply: Ok. Manuscript was modified

Specific Comment 22. P5, Line 17. Report briefly the definition of diameter derived from the microscopy (e.g., equivalent area, longest axis, etc). Reply: Ok. Manuscript was modified. We reported observable mean diameter.

Specific Comment 23. P5, Line 17. Figure 2d reports apparently size in mm, not micrometres. Reply: Ok. Manuscript was modified. Figure 2d was modified. Specific Comment 24. P5, line 18. Rosin-Rambler (abstract) or Rosin-Rammler? Reply: Ok. Manuscript was modified

Specific Comment 25. P5, Line 35. Are these two different references? If so, they should be disambiguated. Reply: Ok. Manuscript was modified. They are two different references.

Specific Comment 26. P6, section 4.2. Whether an accompanying model description paper is available or not, sufficient detail of the modelling approach should be given to allow the reader to understand the model set-up and experiments. Please describe the model discretization and turbulent closure as a minimum. Please provide some justification for the size of domain and for the boundary conditions chosen. Please provide details of model spin up to steady state. Please explain how 1 hour steady state models are to be compared to 24-hour average measurements. Please describe what microphysics, if any, is included in the model. Reply: Ok. Manuscript was modified The information requested about the implementation of the CFD model is included later in the section under the subtitle: Implementation of the NR-CFD model.

We used the standard k-$\varepsilon$ turbulence model

We used a 1500-m-long, 60-m-high and 10 m-depth computational domain. Computational domain dimensions were selected as the minimum required for the boundaries not to interfere with dispersion under the emission conditions studied. Even though the simplicity of the geometry allows 2D simulations, particle dispersion is highly affected by turbulence and therefore 3D simulations are required.

We used a condition of symmetry (zero gradient normal to boundary) at the upper ceiling. Pressure outlet was used as boundary condition at the exit and a periodic boundary condition was used for the lateral walls. On the surface downwind from the road, we considered the Air–particulate matter–ground interaction and used the boundary condition that traps the particles arriving at the surface (ANSYS, 2012a). We expressed the entry of air into the computational domain as a speed profile of a fluid on a flat surface using Equation 6, which describes a neutrally stratified atmospheric boundary layer (Panofsky and Dutton, 1984; Zanneti, 1990). . . . . . .

Steady state vs. transient simulations: Particle dispersion is a natural phenomenon that varies with time. To determine its impacts on human health and the environment, average short-term ($\sim$1 day) and long-term ($\sim$1 year) ground-level concentrations are needed. As the modelling of transient-state particle dispersion via CFD for 1 day and certainly for one year is computationally prohibitive, we simplified the problem by using short-interval modelling, where it could be assumed a steady state condition. In practice, 1-hour intervals are appropriate, as meteorological data are reported in this way.

To obtain pollutant concentrations over extended periods of time, we calculated for each hour i the values of input parameters to the NR-CFD model (average emission rate, wind speed, wind direction). Then, pollutant concentration ($C_{i,j}$) is obtained for each hour and position (j) downwind from the road. Finally, average daily and annual values are obtained for each distance from the road (($C\_j$) ÌĚ). If particulate matter

emission remains constant, average values are obtained by Equation 10 where fk,q is the frequency at which speed Uk, appears in the wind rose for each wind direction (q).

$$C\_j = \sum \sum f\_(k,q)C\_(k,j) \quad (10)$$

For winds flowing in directions other than perpendicular to the road, we maintained the magnitude of wind speed unaffected and computed its contribution to particulate matter concentration at receptor j as if the receptor j were located at an equivalent distance from the road (xe) (Figure 4, Equation 11).

$$x\_e = x / \cos(\theta) \quad (11)$$

Further details on the implementation of the NR-CFD model are reported in (Huertas et al., 2018).

Specific Comment 27. P6, Line 34ff. I don't believe that readers will accept you can model all kinds of vehicle induced aerosol with a single 'quartz' model tracer following a Rosin-Rammler size distribution. It would be much more persuasive to stick to modelling the suspended silt that makes up the vast majority of the mass concentration in the hi-vol samples. Reply: Ok. Manuscript was modified. We deleted all mentions to other types of particles in this section.

Specific Comment 28. P7, Line 9. Finally, we learn that the model is FLUENT, set-up with a variety of standard settings. Please completely re-order the description of the modelling to start with the name of the commercial modelling system and describe the important set-up parameters as asked for above. Reply: Ok. Manuscript was modified.

Specific Comment 29. P8, Line 1. The calibration procedure is not clear. Calibration implies that some parts of the measured data were used to refine model parameters and then the calibrated model used to simulate a different part of the data. Please explain. Reply: Ok. Manuscript was modified. We did that, except that all experimental data was used to adjust the model. Then the calibrated model was used to study dispersion of gases, vertical profiles of concentration etc. We re-wrote the paragraph

to make it clear.

Specific Comment 30. Figure 5b. How long are the long-term averages? Please make all captions self explanatory. Reply: Ok. Manuscript was modified

Specific Comment 31. P8, Line 11. If the measurements are averages please also plot standard deviations (or, better, plot medians and quartiles). Reply: Ok. Manuscript was modified. Figure 5 was modified.

Specific Comment 32. P8, Line 16. Please report numbers using standard scientific notation. RMSE should have associated units. Reply: Ok. Manuscript was modified

Specific Comment 33. P8, Line 26, It is a basic property of the Gaussian plume model that downwind concentrations are proportional to the emission rate, so Figure 6a is not needed. Figure 6b is more interesting, but only if some description of the model is provided that would account for non-linear behaviour with emission rate. Since the concentration further downwind is exactly proportional to emission rate for the CFD simulation, it is more pertinent to ask what is causing the spread near the source. Reply: Ok. Manuscript was modified Figure 6a was deleted It also shows that near the road (x*<1) C* is highly disperse. We believe that the spreading effect is caused by the perturbations that the vertical flow of particles emitted from the road causes to horizontal incoming wind flow. Those mixing flow effects disappear downwind when the mix-flow becomes again uniform and parallel to the ground surface.

Specific Comment 34. Figure 6b and 6d. I am not sure how "zero" can appear on a logarithmic scale. This will be confusing, especially for junior scientists, and should be removed. Reply: Not Ok. Figures 6b and 6d are semi-log plots. On the x-axis we used log scales and they ranges from 0.1 to 1000. On the y-axis we used normal scales and they ranges from 0 to 10.

Specific Comment 35. P8, Line 30ff. It is, again, a standard result from Gaussian plume models that the concentration at a point varies inversely with wind speed. This para

therefore shows what a good job Gaussian plume modelling does of capturing the time-average concentration profile downwind of a source, which has been demonstrated many times before. Again, the behaviour of the CFD for x* <1 is more interesting. Reply: Ok. Manuscript was modified We removed the paragraph from the manuscript.

Specific Comment 36. P9. If deposition is negligible and coagulation and condensational growth are not applicable/ accounted for, then the ratios of size fractions are bound to remain constant. Reply: Yes. That is the point of the subsection entitle PM10 and PM2.5 concentrations downwind

Specific Comment 37. P10, Line 10. It is not intuitive to expect a Gaussian vertical distribution from AERMOD. This would be the case for a chimney but not for a ground source. Reply: Ok. Manuscript was modified. We removed this word from the paragraph.

Specific Comment 38. P10, Line 46ff. Please explain why the model results for gas phase tracers are very much smoother than those for TSP. Reply: Ok. Manuscript was modified.

Specific Comment 39. P11, Line 23. This material should be much earlier on when the concept of 'area affected' is introduced. It is important to state what averaging time is used in the air quality standard you are using, and to compare similar modelled and measured averaging times. Reply: Ok. Manuscript was modified This section is located at the end of the document because the determination of the road impact area is an application of the model.

Specific Comment 40. P12. The conclusions should be re-written in light of the revisions suggested by the referees. Reply: Ok. Manuscript was modified

Specific Comment 41. References. If Huertas and Prato is "in press" please provide the journal name. Reply: Ok. Manuscript was modified

Specific Comment 42. Throughout: please could the font size and line spacing be

made consistent. Reply: Ok. Manuscript was modified

Best Regards Dr. José Ignacio Huertas School of Engineering and Science-EIC Automotive engineering research center- CIMA (http://cima.tol.itesm.mx/) Tecnológico de Monterrey (http://www.itesm.edu/) Phone: (52) 81 8358 2000 Ext 5293

Please also note the supplement to this comment:
https://www.atmos-chem-phys-discuss.net/acp-2017-753/acp-2017-753-AC2-supplement.pdf